# Modelling armed conflict risk under climate change with machine learning and time-series data

Quansheng Ge [1,8], Mengmeng Hao [1,2,8], Fangyu Ding [1,2✉], Dong Jiang [1,2,3✉], Jürgen Scheffran [4], David Helman [5,6] & Tobias Ide [7]

Understanding the risk of armed conflict is essential for promoting peace. Although the relationship between climate variability and armed conflict has been studied by the research community for decades with quantitative and qualitative methods at different spatial and temporal scales, causal linkages at a global scale remain poorly understood. Here we adopt a quantitative modelling framework based on machine learning to infer potential causal linkages from high-frequency time-series data and simulate the risk of armed conflict worldwide from 2000–2015. Our results reveal that the risk of armed conflict is primarily influenced by stable background contexts with complex patterns, followed by climate deviations related covariates. The inferred patterns show that positive temperature deviations or precipitation extremes are associated with increased risk of armed conflict worldwide. Our findings indicate that a better understanding of climate-conflict linkages at the global scale enhances the spatiotemporal modelling capacity for the risk of armed conflict.

[1] Institute of Geographic Sciences and Natural Resources Research, Chinese Academy of Sciences, Beijing 100101, China. [2] College of Resources and Environment, University of Chinese Academy of Sciences, Beijing 100049, China. [3] Key Laboratory of Carrying Capacity Assessment for Resource and Environment, Ministry of Land & Resources, Beijing 100101, China. [4] Institute of Geography, Center for Earth System Research and Sustainability, University of Hamburg, Hamburg 20144, Germany. [5] Institute of Environmental Sciences, Department of Soil and Water Sciences, The Robert H. Smith Faculty of Agriculture, Food & Environment, The Hebrew University of Jerusalem, Rehovot 7610001, Israel. [6] Advanced School for Environmental Studies, The Hebrew University of Jerusalem, Jerusalem, Israel. [7] Centre for Biosecurity and One Health, Harry Butler Institute, Murdoch University, Murdoch, 6150 Perth, WA, Australia. [8] These authors contributed equally: Quansheng Ge, Mengmeng Hao. ✉email: dingfy.17b@igsnrr.ac.cn; jiangd@igsnrr.ac.cn

Armed conflict is the intervention of armed forces resulting from a disagreement over territory, policy and/or resources between two or more organized armed groups, governments or non-governments[1,2]. Among various conceptions of armed conflict most prominent is the Uppsala Conflict Data Program (UCDP) georeferenced event dataset (GED) which defines an armed conflict event as "an incident where armed force was used by an organised actor against another organized actor, or against civilians, resulting in at least 1 direct death at a specific location and a specific date"[3]. This allows to measure the frequency of armed conflict events in terms of incidence (armed conflict event in a particular year) and onset (incidence without armed conflict event in previous year) in spatial and time units (see the equations in Supplementary Information). In our analysis, we count the existence of both incidence and onset of armed conflict events, while other aspects of armed conflict are not specified such as conflict intensity or consequences, conflict parties, historical context or other patterns of conflict which are considered in the literature.

According to UCDP-GED, more than 91,000 armed conflict events occurred globally between 2000 and 2015, which directly caused approximately 654,000 deaths, including nearly 144,000 civilians[4]. In Asia and Africa, the Armed Conflict Location and Event Data Project (ACLED) reported that more than 23,000 armed conflict events occurred from January to August 2017, killing approximately 24,000 people[5]. Although the global trend of armed conflict events has declined in both number and intensity over a decade long perspective, with particularly sharp declines in higher-intensity conflict[6,7], the frequency of armed conflict events in several areas shows an upward trend, becoming more concentrated in Africa, the Middle East and South Asia[5].

In recent years, understanding conflict risk has drawn increased attention from an interdisciplinary group of scientists because it is of great significance for human safety and security[8]. The term conflict risk has been associated with the probability of armed conflict events[9] which is adapted here to refer to the frequency of armed conflict events which involves armed conflict incidence and armed conflict onset. Both researchers and policy makers have recently discussed intensively whether climate change impacts conflict risks[9,10]. The United Nations Security Council, for instance, has conducted discussion on climate change and security in every year since 2018.

Research on the climate–conflict connection covers a wide range of climate phenomena as well as conflict dimensions which makes the diverse outcomes of the different studies difficult to compare. Although scientists have yet to agree on the causal climate–conflict connections[11–14], there is an increasing acceptance that climate change or changes in climate variability increase the risk of armed conflict in certain circumstances[9,15,16]. For instance, the fifth assessment report (AR5) of the Intergovernmental Panel on Climate Change (IPCC) shows that climate change has the potential to increase rivalries between countries over shared resources, meaning that climate change increases the threat of armed conflict[17]. In addition, several studies use archaeological excavation data to address the connection between climate variability and conflict on a long-term or global scale basis. For example, Kuper and Kröpelin linked climatic variations with prehistoric occupations during the past 12,000 years based on radiocarbon data. They demonstrated that the climate-controlled desiccation and expansion of the Saharan desert since the mid-Holocene period may ultimately be considered a driver of Africa's evolution up to modern times[18]. Hsiang et al. conducted a comprehensive review of the literature from 1986 to 2013 on the intertemporal associations between climatic variables and conflict and found that the magnitude of the climate's effect on modern conflict is highly statistically significant[19]. These studies explore the connection between conflict and climate fluctuations based on long-term data across the millennium and cannot avoid the common problem regarding high levels of uncertainty. Compared with these long-term series datasets, climate and armed conflict data have become more accurate over the last thirty years. This development can be highly beneficial for research on climate change and armed conflict[20]. Schleussner et al. employed the event coincidence analysis method to test for statistical interrelationships between climate-related disasters and conflict risk at the global scale for the period from 1980 to 2010. They revealed that approximately 23% of conflict outbreaks in ethnically highly fractionalized countries robustly coincide with climatic calamities[21]. Likewise, Ide et al. find that climate-related disasters increase the likelihood of armed conflict in contexts of ethnic exclusion, low human development and large populations[14]. Bretthauer argues that climate-induced water and land scarcity increase armed conflict incidence[22]. However, the above three studies primarily explored the relationship between climate change and conflict at the national level[14,21,22], which disregards spatial variations in conflict risk within countries and assumes that the conflict's presence is uniform across large areas. In several studies, there is no clear evidence of a connection between climate variability and conflict[12]. For this phenomenon, Uexkull et al. proposed that one reason may be the failure to properly specify the appropriate time and space ranges within which climatic extremes can undermine social stability and increase conflict risk. Therefore, they quantified the drought-conflict relationship on politically relevant ethnic groups and growing-season periods. The results showed that the likelihood of a sustained conflict increased with local droughts between 1989 and 2014 in Asia and Africa[23].

The simulation and prediction of conflict risk at finer scales are essential for promoting societal stability and peace. Scheffran et al. presented a systematic analytical framework for the link between the climate system, natural resources, human security and society stability[24,25]. Hegre et al. outlined a methodological framework and combined several modelling approaches to evaluate conflict risk at the country and subnational level in Africa[26]. However, exploring the causal climate–conflict linkages at the global scale is still a challenging task. In recent years, simulation- and data-driven approaches (named machine learning) have been proven to have the potential to solve many complex problems based on large amounts of data[27–29], including climate-conflict linkages[30]. Therefore, we propose the potentially tractable question of whether the machine-learning approach could be used to discover patterns between conflict risk and high-dimensional covariates.

In this research, we combined a machine-learning approach with high-frequency time-series data to model armed conflict risk under climate change. We proposed a hypothesis that where such patterns exist, a machine-learning model fitted from a single-year dataset should have a certain ability in predicting armed conflict risk in other years with the pattern we capture. We adopted a time-cross validation method to prove our hypothesis at a more detailed scale. We showed that the risk of armed conflict is primarily influenced by stable background contexts with complex patterns, followed by climate deviations related covariates. We further revealed that positive temperature deviations or precipitation extremes are associated with increased risk of armed conflict worldwide. We also simulated the risk of armed conflict worldwide from 2000 to 2015. Thus, this study provides a novel insight into understanding the climate–conflict link at the global scale.

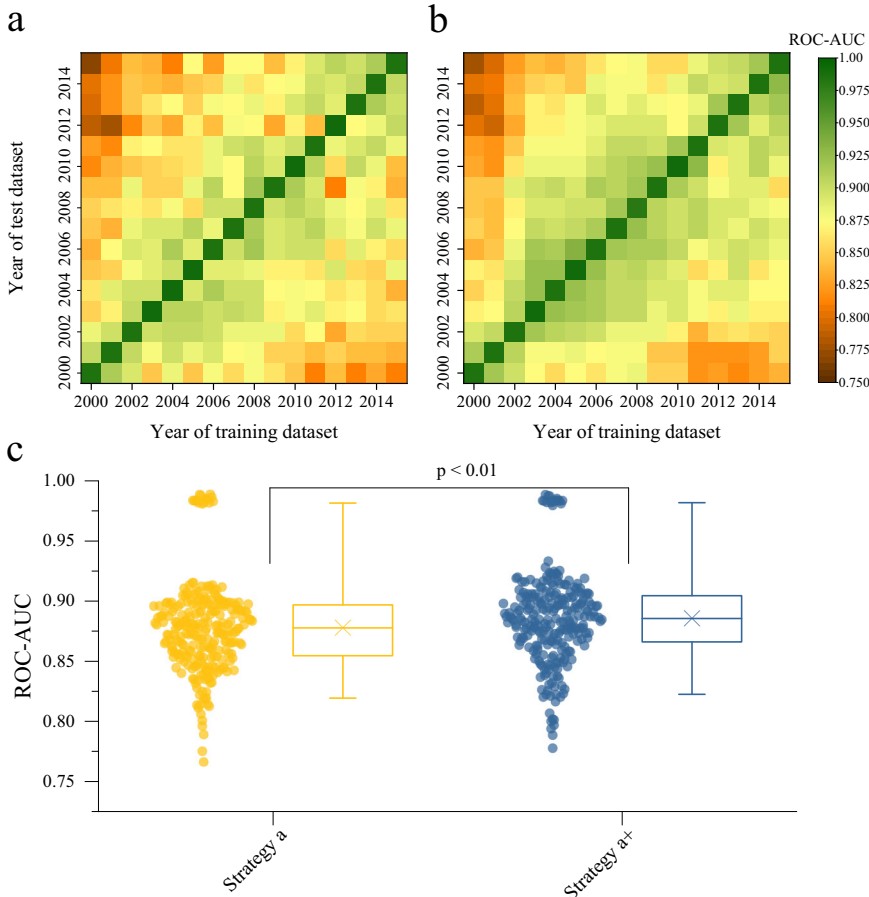

**Fig. 1 Validated performance on a time scale of the boosted regression tree models.** The boosted regression tree (BRT) models were trained on one-year incidence samples under strategies a (**a**) and a+ (**b**). Throughout the validation process, the values of area under the receiver operator characteristic curve (ROC-AUC) range from 0.750 (Dark brown) to 1 (Green). The $p$ value ($p = 0.0013$) was determined by the two-tailed Mann–Whitney test ($n = 256$), representing a comparison (**c**) between strategies a (the pairing of stable background contexts with one-year climate deviation related covariates) and a+ (same as a, but with two-year climate deviations) during the validation process. For each box plot, the '×' indicates the mean; the box indicates the upper and lower quartiles and the whiskers indicate the 5th and 95th percentiles of the data.

## Results

**Time-cross validation analysis.** Based on the UCDP GED and high-frequency time-series covariate dataset, we constructed a series of armed conflict incidence samples and armed conflict onset samples under the four strategies (See Methods). To verify the feasibility of machine learning models, we propose a time-cross validation method in which the boosted regression tree (BRT) models trained on one-year samples should have good performance on the samples of other years. Figure 1 shows the performance of BRT models on the time scale under strategies a and a+, which illustrate that the area under the receiver operator characteristic curve (ROC-AUC) of the 20 ensemble BRT models trained on one-year incidence samples under strategy a+ was higher than that of strategy a ($0.886 \pm 0.039$ s.e. vs $0.878 \pm 0.038$ s.e., $p < 0.01$). The detailed performances of the 20 ensemble BRT models under four strategies during time-cross validation process are shown in Supplementary Figs. 2–5 and Supplementary Tables 1 and 2. A predictor with a ROC-AUC value of 0.5 is a random predictor. Therefore, the ensemble BRT models are positive predictors. The time-cross validation results prove the hypothesis that there is a link between conflict risk and high-dimensional covariates.

**Performance assessment of boosted regression tree models.** In order to avoid the models skewing to the single-year sample, we merged the samples from 2000 to 2015 to train the BRT models.

Based on the 20 simulation processes, pairing stable background contexts with climate variability (strategies a and a+) can simulate the spatial-temporal dynamics of armed conflict incidence well, as evidenced by the 10-fold cross-validation ROC-AUC ($0.937 \pm 0.001$ s.e.). Compared to strategy a, strategy a+ considers 24-month climate deviations from normal, which provided a higher 10-fold cross-validation ROC-AUC ($0.939 \pm 0.002$ s.e.) value. The performance assessment of the 20 ensemble BRT models trained on all samples under four strategies are described in Supplementary Tables 3–6. Illustrated that the significant differences (* means $p < 0.05$) that were observed for the performance of the BRT models trained on all samples under strategy a+ was comparable to those of strategy a.

**The relative contributions of each covariate.** For the 20 ensemble BRT models trained on all incidence samples from strategies a and a+, Supplementary Figs. 6 and 7 reveal that the main predictors are mean temperature ($46.493 \pm 1.187$ s.e. % and $45.944 \pm 1.171$ s.e. %, positive association), natural disaster hotspots ($15.925 \pm 0.725$ s.e. % and $15.706 \pm 0.753$ s.e. %, complex association), mean precipitation ($10.609 \pm 0.885$ s.e. % and $10.545 \pm 0.831$ s.e. %, positive association), socio-economic covariates ($9.758 \pm 0.667$ s.e. % and $9.684 \pm 0.648$ s.e. % for urban accessibility, negative association; $3.150 \pm 0.166$ s.e. % and $3.207 \pm 0.197$ s.e. % for nighttime lights, positive association), elevation ($5.900 \pm 0.342$ s.e. %

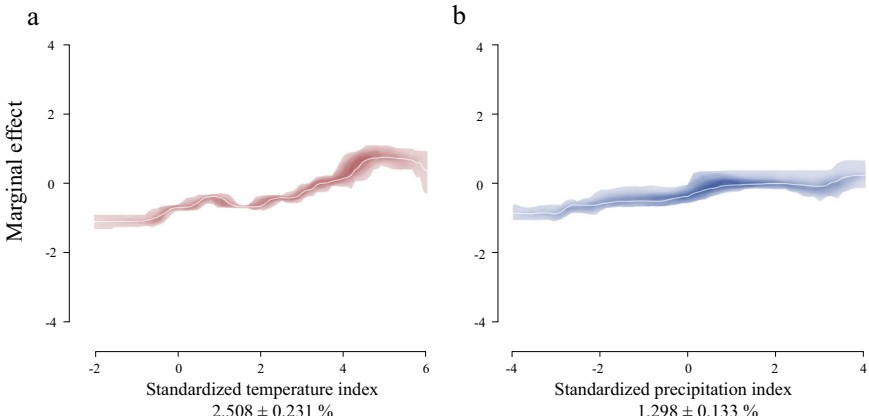

**Fig. 2 Marginal effect curves of each climate deviation related covariate.** The marginal effect curves of (**a**) standardized temperature index and (**b**) standardized precipitation index were generated by the boosted regression tree (BRT) ensemble fitted to the full incidence samples under strategy a+. The white lines represent the mean effect curves calculated from the ensemble BRT models. 95% confidence intervals of climate variables are indicated by color: red, standardized temperature index; blue, standardized precipitation index. Sub-plots are ordered by the mean relative contribution (%) of covariates, with ± standard deviation (%) given within each sub-plot.

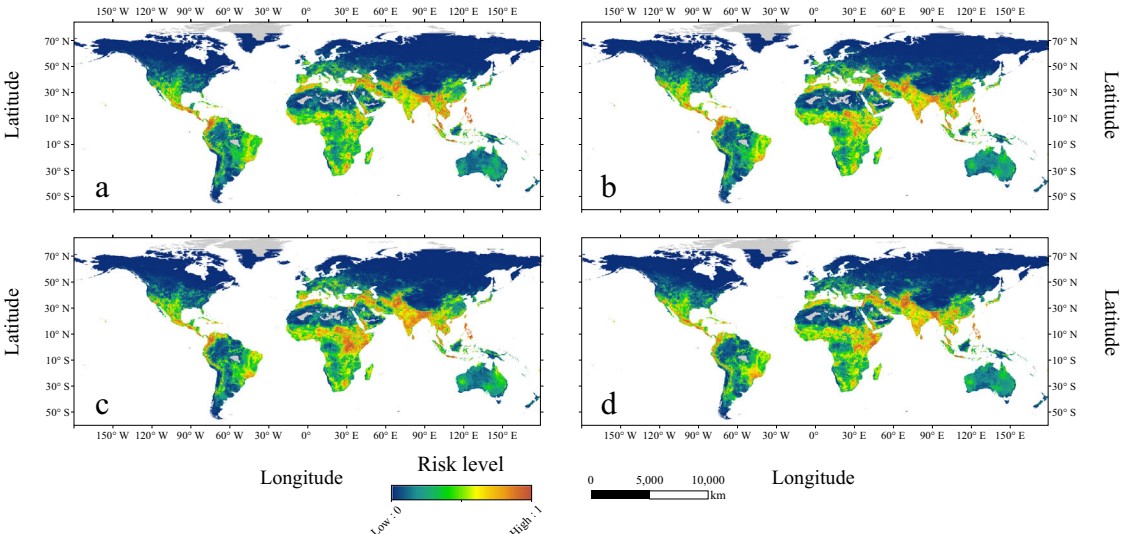

**Fig. 3 Maps of the global simulated risk of armed conflict incidence at 0.1° × 0.1° spatial resolution.** The global risk of armed conflict incidence in (**a**) 2000, (**b**) 2005, (**c**) 2010, and (**d**) 2015 were generated by the 20 ensemble boosted regression tree (BRT) models trained on all incidence samples under strategy a+. The simulated risk level ranges from 0 (blue) to 1 (red), and grey denotes the areas with insufficient data.

and 5.578 ± 0.293 s.e. %, positive association), politically relevant ethnic diversity (2.889 ± 0.191 s.e. % and 2.835 ± 0.189 s.e. %, positive association) and normalized difference vegetation index (2.762 ± 0.197 s.e. % and 2.695 ± 0.190 s.e. %, complex association). Figure 2 and Supplementary Fig. 8 illustrate that the relative contributions of standardized temperature index (1.720 ± 0.167 s.e. %, positive association; 2.508 ± 0.231 s.e. %, positive association) and standardized precipitation index (0.794 ± 0.090 s.e. %, positive association; 1.298 ± 0.133 s.e. %, positive association) are relatively low when compared with stable background contexts for strategies a and a+. The relative contributions of each covariate for the 20 ensemble BRT models trained on all onset samples under strategies a and a+ are shown in Supplementary Figs. 9–12.

**Temporal and spatial distribution of global simulated risk level of armed conflict.** Figure 3 depicts the simulated probability of armed conflict incidence in 2000, 2005, 2010 and 2015 based on the 20 ensemble BRT models trained on all incidence samples under strategy a+, showing the changes among the simulated risk

level derived from different years. In 2000, the simulated high-risk areas for armed conflict incidence were concentrated in Mexico and Central America, northwestern South America (Venezuela and Colombia), Africa (Guinea, Algeria, Uganda, Kenya and northern part of Ethiopia) and several portions of Asia (Iran, Afghanistan, Pakistan, India, Nepal, Bangladesh, Thailand, Malaysia and Philippines) (Fig. 3a). Compared with Fig. 3a, Fig. 3b reveals that the simulated high risk for armed conflict incidence in areas south of North America extends from southern Mexico to northern Mexico. In addition, there is increased risk of armed conflict incidence in eastern Turkey and eastern Afghanistan, while the risks of armed conflict incidence in northern Algeria were reduced in 2005. The main differences between Fig. 3b, c occur in Mexico, Colombia, Morocco, Sudan, Uganda, Afghanistan and India. For example, the simulated risks of armed conflict incidence in southern Sudan, Uganda and eastern India in 2010 were higher than those in 2005, and the risks of armed conflict incidence in northern Mexico and northwestern Colombia in 2010 were lower than those in 2005. In 2015, the risks of armed conflict incidence in northwestern Mexico, Afghanistan and eastern part of Ethiopia were higher than the

simulated results in 2010 (Fig. 3d). The final risk level maps derived from the mean of 20 ensemble BRT models trained on all incidence samples or all onset samples are shown in Supplementary Figs. 13–20, respectively.

## Discussion

Previous studies conducted by O'Loughlin et al. suggested that conflict risk is associated with climate anomalies but is influenced more by political, socioeconomic, and geographic contexts, especially in sub-Saharan Africa[20,31]. Our findings reveal that there are similar patterns at the global scale. For instance, the stable background covariates (see Supplementary Information) greatly contributed to the spatial-temporal distribution of armed conflict incidence with a mean relative contribution of more than 96.0% (Supplementary Table 7). Compared with the stable background covariates, the standardized temperature index or standardized precipitation index had relatively little effect on the simulated results, but covariates related to climate deviation cumulatively accounted for more than 2.5% of the simulated results. This is the reason for why the simulated risk level of armed conflict incidence in the local regions varies in different years. We interpret this as evidence of an impact of climate change on conflict risk. Supplementary Table 3 suggests that considering the two-year climate deviations can slightly improve the performance of the BRT models. This result can be seen as partially supporting other research findings that sequential multi-year deviations from normal climate conditions (to which societies are adapted) may in-part or by-whole affect the stability of societies, both historically[32,33] and in the current time period[34]. Supplementary Table 7 reveals that the long time period climate deviations have a greater impact on risk level with the relative contribution values of 3.806%. This is in the lower range of, but in line with the diverse disciplines experts' judgments that 3–20% of conflict risk are related to climate change[9].

While an increasing number of quantitative studies find that climate change has an impact on armed conflict incidence, evidence regarding climate change and armed conflict onset is more scarce and contested[10,23,35]. Our study thus not only simulates the likelihood of armed conflict incidence, but also further explores the feasibility of simulating armed conflict onset. Based on the definitions adopted by van Weezel[36], we constructed an incidence and an onset indicator to represent conflict risk and carried out modelling analysis separately. The findings further suggest that combining machine learning with high-frequency time-series data has great potential in predicting the risk of armed conflict onset at a global scale (Supplementary Figs. 4, 17 and 18). In addition, our results also indicate that armed conflict onset is more sensitive to climate change than armed conflict incidence at a global scale, as shown in Supplementary Tables 7 and 8.

Our procedure allows for quantifying the relationship between covariates and armed conflict at a global scale. Overall, the discovered patterns derived from a large amount of data are complex. This is the case because different meteorologic, geographic, political and socioeconomic contexts may make human beings adapt differently to environment stress[37,38], leading to varying social stability responses to climate change. However, there are several universal patterns, as shown in Supplementary Figs. 6, 7, 9 and 10. For instance, the positive association between conflict risk level and ethnic diversity illustrates that a greater diversity of the politically relevant ethnicity leads to higher risk of conflict, which is consistent with several previous studies[21,39–41]. Meanwhile, there is a positive link between conflict risk level and urban accessibility, revealing that transportation hubs can easily become the outbreak site of conflict since they play a key role for controlling territories and conflict logistics[42,43]. For the climate deviations related covariates, a few studies have suggested that negative temperature deviation in temperate locations may lead to various forms of conflict and negative precipitation deviation coincided with social instability[23,44–46]. However, modern humans' adaptability to climatic changes is much higher than that recorded in historical studies due to the improvement in technological adaptability and the increase in complexity of social structure. It is still likely, however, that climate change exceeds the adaptive capacities of specific regions (e.g., when they are remote and agriculturally dependent) or groups (e.g., when they are poor and politically excluded). This makes inferences drawn from singular cases (e.g., the Syrian civil war) for the global scale problematic. Figure 2 and Supplementary Figs. 8, 11 and 12 indicate that positive temperature deviations or precipitation extremes are associated with increased risk of armed conflict across the globe from 2000 to 2015. This vindicates the results of other studies, such as those of Hsiang et al.[19], Mach et al.[9], and Helman and Zaitchik[47]. In addition, our findings reveal that rising temperature has a greater nonlinear impact on the risks of armed conflict incidence and armed conflict onset than precipitation deviation at a global scale.

Based on high-dimensional datasets and large volumes of occurrence records, we used the BRT models to simulate the global risks of armed conflict incidence and armed conflict onset at a grid-year level (0.1° × 0.1°) under four strategies. Globally, the distribution of conflict risk from 2000 to 2015 shows obvious spatial agglomeration characteristics, which can be well simulated by the models. The simulated results depend on the distributions of the samples. To improve the simulation accuracy and reduce the impact of low-risk samples, we repeat the process of randomly selecting the low-risk samples 20 times and constructing the BRT models based on each sample set. The maps of uncertain level associated with these simulations are generated based on standard deviation values calculated for each grid across the 20 ensemble BRT models, which are presented in Supplementary Figs. 21–28, respectively. The uncertain level maps illustrate that there is low simulation uncertainty.

In this study, there are some caveats. First, media reports represent one source of data for UCDP GED, and well-known media bias may add uncertainty to our results to some extent. Although several measures (i.e., triple-checked) were employed to ensure high quality of final dataset[4], UCDP was unable to resolve the bias in the GED completely and include all armed conflict events in its dataset. Second, our analysis was based on the global-scale multi-dimensional spatial-temporal refined dataset. Due to the lack of refined datasets of cultural and historical factors, our training of the machine learning models is limited in quantifying the role for these variables. However, with more samples for machine learning models to train on, our global-scale refined analysis can help models capture more reliable relationships. Third, there is no general theory to explain the causal mechanism of the climate–conflict link at the global scale, but our modelling framework may be helpful for early warning of conflict risk. This is indicated by the comparative analysis that the predicted risks of armed conflict incidence in Africa (Supplementary Fig. 29b) are generally consistent with the risk level estimated by Hegre et al.[26]. In the aggregate, our study provides a better understanding of a climate–conflict link at the global scale and enhances the spatial-temporal modelling capacity for the risk of armed conflict worldwide.

## Methods

**Analysis**. We assume that a machine learning model should be able to infer potential patterns between armed conflict and climate variability based on

established facts, which may help to simulate the risk of armed conflict. The potential patterns may be complex. To capture complex responses, the boosted regression tree (BRT) modelling framework was adopted based on the R version 3.3.3 64-bit statistical computing platform. In the present study, independent variables are classified into two categories: stable background contexts and climate deviation related factors. The former is used to reflect various meteorological, geographical, political and socioeconomic contexts, while the latter is adopted to depict the extent of climate change. Based on UCDP GED, two binary dependent variables, including armed conflict incidence and armed conflict onset, were defined for each $0.1° \times 0.1°$ grid on a yearly basis to represent armed conflict risk. If there are one or more instances of armed conflict event in one grid in a single year, the armed conflict incidence indicator is coded as one (high-risk) for the grid. In addition, if a new armed conflict event outbreak occurs after one calendar year of inactivity in one grid, armed conflict onset is assigned the value of one for the grid. Both binary dependent variables are otherwise assigned the value of zero (low-risk). For each year, an equivalent amount of low-risk samples and high-risk samples are randomly selected to construct the one-year samples and to train the BRT models. The detailed description of the BRT modelling framework, of independent variables and dependent variables can be found in the Supplementary Information.

In this study, we use the term conflict risk which has been associated with the probability of armed conflict events[9], adapted here to the frequency of armed conflict events according to the UCDP GED database (see Introduction). This allows to measure the incidence and onset of armed conflict in spatial and time units (Supplementary Information). We design two strategies to construct the sample's dimensional information: (a) the pairing of stable background contexts with climate deviation related covariates, and (b) combining long-term precipitation and mean temperature distribution with the climate deviation related covariates. In addition, we add two strategies named a+ and b+ to analyze the effects of longer-term climate deviations. More information about these four strategies are described in Supplementary Information. All covariate layers are aggregated for each year from 2000–2015 to a common 0.1° global grid using a unified coordinate system (i.e., WGS-84). This results in a total of 1,443,579 grid units after excluding grids with missing covariates data. The analytic process overview is shown in Supplementary Fig. 1.

## Data

*Armed conflict database*. GED 17.1 version data are taken from the UCDP website, which is an openly available armed conflict dataset with georeferenced information[4]. This dataset records three types of armed conflict events (state-based conflict, non-state conflict and one-sided violence) with at least 1 direct death at a specific location and with specific data. The maximum spatial resolution of the UCDP GED 17.1 version is the individual village or town. Therefore, we can localize armed conflict to $0.1° \times 0.1°$ grid based on latitude and longitude coordinates.

*Precipitation*. Studies find that deviations from normal precipitation systematically increase the risk of various forms of conflict, which is apparent across time periods spanning 10,000 BCE to the present and across all major world regions[19,21]. For instance, a local negative precipitation deviation is found to increase conflict risk especially for agriculturally dependent regions[23]. Given the ability to broadly reflect the drought/humid episodes across the globe[48], the standardized precipitation index is adopted to indicate deviation from the historical average precipitation (since 1970). The Climate Research Unit TS4.0 global dataset downloaded from the Climatic Research Unit (CRU) of University of East Anglia was used to construct a monthly gridded land surface precipitation dataset on $0.5° \times 0.5°$ grids for the period from 1901 to 2015. Then the monthly precipitation dataset was used to generate long-term (1970–1999) mean precipitation distribution data, one-year and two-year standardized precipitation index (2000–2015). To match other covariates, we resampled these data to $0.1° \times 0.1°$ grids.

*Temperature*. Temperature variability has been linked to conflict[49]. For example, colder temperatures in temperate climates resulted in crop failure, and warmer deviations introduced agricultural stress in warmer climates[20]. In the present study, the standardized temperature index was used to measure the deviation from the long-term temperature (since 1970). From the CRU website, we acquired the monthly mean temperature dataset that is arithmetically derived from the Climate Research Unit TS4.0 global dataset at each 0.5° latitude/longitude grid cell across the global land surface. Then long-term (1970–1999) mean temperature distribution data, and the one-year and two-year standardized temperature index (2000–2015) were produced and resampled to $0.1° \times 0.1°$ grids.

*Vegetation index*. Previous literature has also illustrated a link between vegetation index and conflict[20]. For instance, the vegetation index could be used to measure food availability for both animals and humans[20]. In this study, we acquired an advanced very high-resolution radiometer (AVHRR) normalized difference vegetation index dataset with an $8 \times 8$ km spatial resolution and a 15-day interval temporal resolution for 1982 through 2015 from the Global Inventory Modelling and Mapping Studies group. Based on the AVHRR dataset, we produce the mean

normalized difference vegetation index layer with a $0.1° \times 0.1°$ spatial resolution to reflect the long-term average level of food availability.

*Natural disaster hotspots*. Based on the event coincidence analysis, there is a coincidence rate of 9% between the occurrence of disasters resulting from natural hazards and the outbreak of armed conflict for the period from 1980 to 2010 at the global scale[21]. Given that different natural disasters may have different impacts on human life, we assume that there are some unexplored relationships between natural disaster hotspots and armed conflict. In the present study, the global multi-hazard frequency and distribution data downloaded from the Socioeconomic Data and Applications Center of Columbia University are employed to present natural disaster hotspots using a simple multi-hazard index.

*Topography*. The role of terrain in military affairs has been discussed in most studies regarding the anatomy of organizations in conflict[50], such as Sun Tzu's Art of War compiled more than two thousand years ago[51]. There is also a well-established link among topography, extreme weather (i.e., hail incidence) and the military[52,53]. Given these situations, we use the elevation dataset obtained from the NASA Shuttle Radar Topographic Mission to depict the topography and adopted this covariate as one of the inputs to the model.

*Ethnic diversity*. The previous literature has shown a close relationship between politically relevant ethnic groups and the risk of armed conflict in large parts of developing countries[39,40]. Ethnic diversity might play a prominent role in conflict-prone regions (particularly in Africa and Asia), thus serving as a predetermined conflict line[21]. In addition, most contemporary civil wars are found in the vicinity of ethnic lines[41]. The Geo-referencing Ethnic Power Relations (GeoEPR) 2014 dataset is adopted in the present research. This assigns every politically relevant ethnic group to vector polygons and provides their distributions on digital maps[54]. Based on the GeoEPR 2014 dataset, we produce a global ethnic diversity layer with $0.1° \times 0.1°$ spatial resolution, which counts the number of different politically relevant ethnic groups in each grid.

*Urban accessibility*. Due to the key role in control territories, the transportation hub is often the target place that strategists must contend for[42]. Previous research conducted by Hegre et al. regarded the travel time to the nearest city as a social covariate in the conflict risk modelling process[26]. In this study, the travel time to the nearest city with a population of 50,000 people or more is used to simultaneously account for urban accessibility. The urban accessibility dataset is open access and downloaded from the European Commission Joint Research Center website, which is derived from land and water-based transportation networks based on a friction-of-distance algorithm[55].

*Nighttime lights*. A recent expert assessment of the relationship between climate change and armed conflict shows that several experts from diverse disciplines regard low socioeconomic development as the key predictor of conflict[9]. In this study, the nighttime lights dataset was used, which could provide a potentially global measure of the changes in socioeconomic development dynamics at certain levels of gross domestic product (GDP) and population-driven growth[56]. In contrast to GDP and other indicators of macroeconomic performance, nighttime lights can reflect the socioeconomic level at an approximately 1 km spatial resolution, which is less immediately affected by commodity price fluctuations. Nightlight data are also increasingly used in climate security research[57]. The stable type nighttime lights data from 2000 to 2013 were downloaded from the website of the Defense Meteorological Satellites Program/Operational Linescan System. Based on the 14-year nighttime lights dataset, we calculate the mean nighttime lights value for each grid cell and then resample the average annual nighttime lights layer to $0.1° \times 0.1°$ grids.

**Reporting summary**. Further information on research design is available in the Nature Research Reporting Summary linked to this article.

## Data availability

All data used in this study is publicly available. UCDP GED 17.1 version can be accessed from https://pcr.uu.se/research/ucdp/. The Climate Research Unit TS4.0 global dataset can be accessed from https://www.uea.ac.uk/web/groups-and-centres/climatic-research-unit/. The AVHRR normalized difference vegetation index dataset can be accessed from https://ecocast.arc.nasa.gov/data/pub/gimms/. Natural Disaster Hotspots dataset can be accessed from https://sedac.ciesin.columbia.edu/. Elevation dataset can be accessed from https://eospso.gsfc.nasa.gov/missions/shuttle-radar-topography-mission. The GeoEPR 2014 dataset can be accessed from https://icr.ethz.ch/. The urban accessibility dataset is available on https://forobs.jrc.ec.europa.eu/products/gam/. Nighttime lights dataset was downloaded from https://ngdc.noaa.gov/.

## Code availability

The codes used in the present study are freely available online at https://github.com/Celyon/ConflictRisk.

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

## Acknowledgements

We sincerely thank Qiaoling Zhu, William Jefferson, Wenxiang Wu, Tian Ma, Yushu Qian and Qian Wang for providing valuable suggestions. This work was funded by the Strategic Priority Research Program of the Chinese Academy of Sciences (Grant No. XDA19040305) and National Natural Science Foundation of China (Grant No. 42001238). The funders of the study had no role in study design, data collection, data analysis, data interpretation, in preparing the paper or in our decision to publish. One of the researchers (J.S.) acknowledges the contribution of this research to the CLICCS Cluster of Excellence funded by the German Research Foundation (DFG).

## Author contributions

Q.G., F.D., and D.J. conceived and designed the study. F.D. collected the data and carried out the computations. F.D. and D.J. analyzed the data. F.D., D.J., and Q.G. wrote the paper. M.H., J.S., D.H., and T.I. gave some useful suggestions to this work. F.D. and D.J. revised the manuscript. All authors critically reviewed the manuscript and approved the final manuscript.

## Competing interests

The authors declare no competing interests.
