## [Peer Review File · Nature Communications]

Modelling armed conflict risk under climate change with machine learning and high-frequency time-series dataREVIEWER COMMENTS

Reviewer #1 (Remarks to the Author):

This paper joins a growing and important literature on high spatial and temporal conflict prediction. While some of this literatures is cited, one set of innovative and influential projects are unrecognized in the current draft. The Violence Early Warning System Project by Hegre, et al (2019, 2020) both uses the same fine-grained conflict event data source but has built a pipeline to accelerate the ingestion of conflict information from news sources over time. While limited to Africa, this pre-existing project also has a set of geographic and political themed models that constitute their ensemble predictions. This is relevant because the three claims to novelty in this paper, a) high spatial resolution, b) spatial clustering, and c) the interaction between low frequency background conditions and higher frequency indicators of conflict, are all included to that previously published project. Because of this prior work, we already have significant evidence of what the authors here term "spatial agglomeration". Ongoing work is attempting to move beyond these spatial clusters, since at high resolution it is relatively easy to see the mass of conflict in a given nearby time or location. It has been much more difficult to predict spatially disjoint onsets of conflict that are not proximate to past conflict history. This fact can be seen when different ensemble and constituent models are compared, background factors like conflict history perform well, with only incremental improvements from other factors. None of the approaches or findings herein extend the existing research. The one new extension that is not stressed in the current paper is the combination of global coverage with high spatial resolution. However, it is never argued why this is an important step of understanding or forecasting conflict.

This does not imply that the current work is not crucial. An explicit comparison to existing models with coherent findings would be a nice contribution to the literature. In that case, a more careful analysis of the ViEWS model with the current approach specifically within Africa could improve the influence of the research. For example, it could be the case that training models on global events improves predictions for Africa. Alternatively, as the results here appear to show, it might be opposite, where regions have distinct patterns and then averaging over long-range dependencies reduces the usefulness of the computed model representations. Given that stable covariates could result from many different causal factors, it is unclear how any interpretation of the usefulness of climate change for understanding new conflicts is made. A potentially more convincing next step could be to first deploy a high performing model of conflict from the literature and train models and then compare true forecasts with that information to a climate-based model/ensemble (or several following the different training schemes herein). A smaller point is that given the rarity of violence in the data, accuracy and AUC are not necessarily the most useful measures given it takes low skill (just choose the majority class) to record high scores in absolute terms. Including baselines from other systems can help provide a relative skill score for comparison. In addition, F1 and the area under precision-recall curves could be helpful in the main paper. It was also unclear to me what the benefit of training on data from the future and testing on the past was in this set up. I can imagine scenarios where this makes sense, but I did not see the research design justification here.

Reviewer #2 (Remarks to the Author):

Armed conflict often occurs in the regions where a disagreement on territory, natural resources and/or culture exists between two or more organized armed groups, governments or non-governments. It is of importance to quantify the risk of an armed conflict in a finer spatial scale for promoting peace and human security. Combined the data on armed conflicts are taken from the UCDP website and climate change data, this study proposed a machine learning-based approach to simulate the global armed conflict risk at $0.1^\circ \times 0.1^\circ$ spatial resolution. A clear insight view on global conflict risk will help promoting societal stability and peace. The study is novel and the results be of interest to others in the community and the wider field. However, the current version of the manuscript does not meet the requirement to be accepted for publication.

My major concerns are as follows:

1. The manuscript tried to reach three objectives: "a) Select a suitable machine-learning algorithm to answer b) If these patterns exist, mine the pattern. c) Simulate and predict the risk of

armed conflicts on a global scale and quantify the uncertainty of the simulation." In the manuscript, authors mentioned " To capture the complex responses, we elected to build several BRT models". It looks that the authors directly selected BRT as the model for study, but we cannot see anywhere to discuss "how to select a suitable machine-learning algorithm". For objective b, it looks not an objective, if so, authors should propose a hypothesis to test it.

2. The covariate data were not clearly introduced why those data and parameters are related to the study topic and how to be selected. Also predictor variables selected may need rigorous statistical tests, such as collinearity test.

3. Types of armed conflict events cannot elaborate leading factors in line with characteristics of natural conditions in target region.

4. In addition, the parameters of the Boosted Regression Tree model were not listed in detail. Thus, supplements is highly recommended.

5. In the introduction, authors indicated that "exploring the underlying mechanisms is still a challenging task.", but no detail discussion can be seen in the section of Discussion.

6. English needs editing and proofreading.

Reviewer #3 (Remarks to the Author):

The paper proposes a machine learning-based approach to simulate the global armed conflict risk at $0.1^\circ \times 0.1^\circ$ spatial resolution for the period from 2000-2015.

The paper must be improved in the following aspects:

(i) The literature review needs to be updated with more papers published in this three last years, since machine learning is a hot research topic

(ii) The authors must provide a flowchart describing the used methodology

(iii) The Table 1 (describing the relative importance of covariates in predicting the global risk of armed conflict events based on the 20 ensemble BRT models trained on all samples from period 2000-2015) must be enriched and verified with knowledge domains from experts.

Responses to reviewers

NCOMMS-20-07482

Title: Modelling armed conflict events with machine learning and high-frequency time-series data

We sincerely thank all the reviewers for their critical reading and constructive comments and suggestions for improving our manuscript. We have carefully addressed all the comments and questions. Answers are typed in **blue**. Following the reviewers' comments and suggestions, we have revised the manuscript accordingly, as detailed below.

Reviewer #1 (Remarks to the Author):

Point No.1 This paper joins a growing and important literature on high spatial and temporal conflict prediction. While some of this literature is cited, one set of innovative and influential projects are unrecognized in the current draft. The Violence Early Warning System Project by Hegre, et al. (2019, 2020) both use the same fine-grained conflict event data source but has built a pipeline to accelerate the ingestion of conflict information from news sources over time. While limited to Africa, this pre-existing project also has a set of geographic and political themed models that constitute their ensemble predictions. This is relevant because the three claims to novelty in this paper, a) high spatial resolution, b) spatial clustering, and c) the interaction between low frequency background conditions and higher frequency indicators of conflict, are all included to that previously published project. Because of this prior work, we already have significant evidence of what the authors here term “spatial agglomeration”. Ongoing work is attempting to move beyond these spatial clusters, since at high resolution it is relatively easy to see the mass of conflict in a given nearby time or location. It has been much more difficult to predict spatially disjoint onsets of conflict that are not proximate to past conflict history. This fact can be seen when different ensemble and constituent models are compared, background factors like conflict history perform well, with only incremental improvements from other factors. None of the approaches or findings herein extend the existing research. The one new extension that is not stressed in the current paper is the combination of global coverage with high spatial resolution. However, it is never argued why this is an important step of understanding or forecasting conflict. This does not imply that the current work is not crucial. An explicit comparison to existing models

with coherent findings would be a nice contribution to the literature. In that case, a more careful analysis of the ViEWS model with the current approach specifically within Africa could improve the influence of the research. For example, it could be the case that training models on global events improves predictions for Africa. Alternatively, as the results here appear to show, it might be opposite, where regions have distinct patterns and then averaging over long-range dependencies reduces the usefulness of the computed model representations. Given that stable covariates could result from many different causal factors, it is unclear how any interpretation of the usefulness of climate change for understanding new conflicts is made. A potentially more convincing next step could be to first deploy a high performing model of conflict from the literature and train models and then compare true forecasts with that information to a climate-based model/ensemble (or several following the different training schemes herein). A smaller point is that given the rarity of violence in the data, accuracy and AUC are not necessarily the most useful measures given it takes low skill (just choose the majority class) to record high scores in absolute terms. Including baselines from other systems can help provide a relative skill score for comparison. In addition, F1 and the area under precision-recall curves could be helpful in the main paper. It was also unclear to me what the benefit of training on data from the future and testing on the past was in this set up. I can imagine scenarios where this makes sense, but I did not see the research design justification here.

Response to point No.1: Thanks for your nice and valuable comments. According to your suggestions, we rephrased the corresponding sentences and added some analysis.

Firstly, we searched for related works conducted by Hegre et al. from 2015-2020 through Google Scholar. We found literature about The Violence Early Warning System Project, which outlined a methodological framework, and read it (Ref. 21) carefully. It combined conflict information derived from news sources with several approaches to modelling the risk of conflict at the country and subnational level in Africa. To highlight important milestones from the analytical framework (Refs. 19 and 20) to the practical stage, we cited The Violence Early Warning System Project conducted by Hegre et al. in the revised introduction section, as follows:

“Hegre et al. outlined a methodological framework and combined several modelling approaches to evaluate conflict at the country and subnational level in Africa (Ref. 21)” (Page 5, lines 111-113).

Secondly, we found full source code of OpenViEWS on the website of GitHub

(<https://github.com/UppsalaConflictDataProgram/OpenViEWS>). According to its section of Ref. 21, we wanted to make a more careful analysis of the ViEWS model with the current approach. The ViEWS model involves many modules which I tried to build and run. Unfortunately, non had turned out to be successful. To facilitate comparison with the fine-grained results derived from the ViEWS model, the risk maps of armed conflicts in Africa in 2018 were generated from 20 ensemble BRT models trained on all samples from 2000-2015 under the four strategies, as shown in Fig. S1. In the aggregate, the risk of armed conflicts has obvious spatial agglomeration characteristics in Africa in 2018. For instance, Fig. S1b reveals that the highest risk level mainly distributed in northern Nigeria, Rwanda, Burundi, Uganda, western parts of Kenya, and Somalia in 2018. From the visual point of view, although there are some differences between several local regions (i.e., Tunisia), Fig. S1b is generally consistent with the results derived from The ViEWS model.

Fig. S1. Maps of the simulated risk of armed conflict in Africa in 2018 are generated from 20

ensemble BRT models fitted from full dataset under strategies (a) a, (b) a+, (c) b, and (d) b+. The simulated risk level ranges from 0 (blue) to 1 (red), and the grey part denotes the areas with insufficient data.

Thirdly, we use machine learning models to explore the underlying patterns hidden in large amounts of data. Although to simulate the risk of armed conflicts at a global scale remains a challenging task for the intricacy of the underlying mechanisms, this still provides us a better understanding of the relationships between conflict and covariates. We can also quantify the risk of armed conflicts with these machine learning models fitted from observational data, which is quite proactive about human security.

Fourthly, the time-cross validation method was used to prove whether the machine-learning approach could discover the patterns between conflicts and high-dimensional covariates. The results that the ensemble models fitted from the past data have a high performance on the future data or the models fitted from the future data have a high performance on the past data illustrate there are similar patterns in the past and future.

Last but not least, in Supplementary Information, area under the receiver operator characteristic curve (ROC-AUC), area under the precision recall curves (PR-AUC) and F_1 -score were adopted as accuracy evaluation indexes (Pages 3-4, lines 35-37 in Supplementary Information; Pages 11-12, lines 146-153 in Supplementary Information; Page 18, lines 213-221 in Supplementary Information), as shown in Supplementary Figs 2-3 and Supplementary Tables 1-2.

Supplementary Figure 2. Validation performance on a time scale of the BRT models trained on one-year samples. Validation performance of strategies a and a+ are shown in the left and right columns, respectively.

Supplementary Figure 3. Validation performance on a time scale of the BRT models trained on one-year samples. Validation performance of strategies b and b+ are shown in the left and right columns, respectively.

Supplementary Table 1. The performance of the 20 ensemble BRT models during time-cross validation process.

Performance	Strategy a		Strategy a+		Strategy b		Strategy b+	
	Mean	Standard Deviation	Mean	Standard Deviation	Mean	Standard Deviation	Mean	Standard Deviation
ROC-AUC	0.878	0.038	0.886	0.039	0.784	0.062	0.798	0.061
PR-AUC	0.851	0.048	0.860	0.049	0.731	0.078	0.751	0.077
F ₁ -score	0.756	0.076	0.767	0.078	0.638	0.105	0.657	0.105

Supplementary Table 2. The performance of the 20 ensemble BRT models trained on all samples under different strategies.

Performance	Strategy a		Strategy a+		Strategy b		Strategy b+	
	Mean	Standard Deviation	Mean	Standard Deviation	Mean	Standard Deviation	Mean	Standard Deviation
ROC-AUC	0.937	0.001	0.939	0.002	0.886	0.003	0.891	0.002

PR-AUC	0.935	0.002	0.937	0.002	0.880	0.003	0.887	0.002
F ₁ -score	0.879	0.002	0.882	0.002	0.827	0.003	0.830	0.003

Reviewer #2 (Remarks to the Author):

Armed conflict often occurs in the regions where a disagreement on territory, natural resources and/or culture exists between two or more organized armed groups, governments or non-governments. It is of importance to quantify the risk of an armed conflict in a finer spatial scale for promoting peace and human security. Combined the data on armed conflicts are taken from the UCDP website and climate change data, this study proposed a machine learning-based approach to simulate the global armed conflict risk at 0.1°×0.1° spatial resolution. A clear insight view on global conflict risk will help promoting societal stability and peace. The study is novel and the results be of interest to others in the community and the wider field. However, the current version of the manuscript does not meet the requirement to be accepted for publication.

My major concerns are as follows:

Point No.1 The manuscript tried to reach three objectives: "a) Select a suitable machine-learning algorithm to answer b) If these patterns exist, mine the pattern. c) Simulate and predict the risk of armed conflicts on a global scale and quantify the uncertainty of the simulation." In the manuscript, authors mentioned " To capture the complex responses, we elected to build several BRT models". It looks that the authors directly selected BRT as the model for study, but we cannot see anywhere to discuss "how to select a suitable machine-learning algorithm". For objective b, it looks not an objective, if so, authors should propose a hypothesis to test it.

Response to Reviewer's comment No. 1: Thanks for your nice suggestions. Indeed, the objective b does not look like an objective. According to your valuable suggestion, we rechecked the corresponding statements and found that the expression and the logic need some improvement. In the Methods section, we have rephrased the sentences:

"We assume that a machine learning model should be able to infer potential patterns between the occurrence of conflicts and climate variability based on established facts, which may help to simulate the risks of armed conflicts. The potential patterns may be complex. To capture the complex responses, boosted regression tree (BRT) modelling framework was adopted based on the R version 3.3.3 64-bit statistical computing platform. The detailed description of BRT

modelling framework can be found elsewhere (Supplementary Information)” (Pages 17-18, lines 335-341).

Boosted Regression Trees section is added in Supplementary Information to describe the detailed information of BRT modelling framework, as following:

“A comprehensive comparison of 16 modelling methods conducted by Elith et al. (2006) (Supplementary Ref. 1) revealed that boosted regression trees (BRT) and maximum entropy mode (Maxent) performed better than other modelling methods. Whilst broadly comparable, BRT tend to out-perform than Maxent at capturing the complex relationships based on a large amount of data. Thus, BRT modelling framework was adopted in the present study.

The BRT model can be described using the following functional forms [1] and [2]:

$$f_t(X) = f_{t-1}(X) + \lambda \cdot \rho_t h(X; a_t) \quad \lambda \in (0,1] \quad [1]$$

$$L(y, f(X)) = \log(1 + \exp(-2yf(X))) \quad [2]$$

where $X = \{x_1, x_2, \dots, x_n\}$ represents stable background contexts and climate deviations related covariates, y is conflict risk, $f_t(X)$ refers to the estimated mapping function from X to y during the t -th iteration process, λ is the learning-rate parameter, ρ_t is the weight parameter, $h(X; a_t)$ is the function of an individual tree, and a_t defines the split variables. During the modelling process, the parameters ρ_t and a_t were estimated by minimizing a binomial loss function (equation [2]).

In the present study, we combined the R version 3.3.3 64-bit statistical computing platform with the extension packages (i.e., dismo and gbm) to build BRT modelling framework and assess the performance. To enhance the robustness of simulation, we created an ensemble of 20 BRT models to generate the conflict risk map using the mean method. Area under the receiver operator characteristic curve (ROC-AUC) (Supplementary Ref. 2-4), area under the precision recall curves (PR-AUC), and F_1 -score (Supplementary Ref. 5) were adopted as accuracy evaluation indexes” (Pages 3-4, lines 17-37 in Supplementary Information).

We have rephrased several statements in the revised introduction. For instance, “a) propose a hypothesis, if these patterns exist, a machine-learning model fitted from a single-year dataset should have a certain predictive ability in other years. b) adopt a formal machine-learning modelling framework to prove the hypothesis at a finer scale by combining the algorithm with a

spatial-temporal refined dataset, both for the dependent variable (armed conflict events) and independent variables (contexts and climate deviations related covariates). c) Simulate the risk of armed conflicts at a global scale and quantify the uncertainty of the simulation.” (Page 6, lines 120-129). Meantime, in the revised version, the sentence, “The time-cross validation results prove the hypothesis that the patterns between conflicts and high-dimensional covariates exist”, was added in the first paragraph of the result section (Page 8, lines 157-158). In addition, the selection basis and detailed information of the boosted regression tree models are added in the Supplementary Information (Pages 3-4, lines 17-37 in Supplementary Information).

Point No.2 The covariate data were not clearly introduced why those data and parameters are related to the study topic and how to be selected.

Response to Reviewer’s comment No. 2: Thanks for your nice comments. According to your valuable suggestion, we rechecked the statements about the covariate data and parameters that are related to the study topic and how they were selected. After that, we found several covariates, including drought index, vegetation, natural disaster hotspots, topography, urban accessibility, and nighttime lights, were not clearly introduced due to inappropriate expression or missing literature. In the revised version, we have made some modifications to the above issues. For instance, when we introduce the nighttime lights dataset, we firstly cited a research conducted by Mach et al. (2019) to emphasize the relationship between socioeconomic development covariates and conflict. Then we made description about the role of night lighting in quantifying social and economic development along with its advantages, which are the reasons why we select the dataset. For urban accessibility, we cited previous literature finished by Zhukov et al. (2012) to reveal the importance of the transportation hub and how it was linked to conflict. We also give an example in which the travel time to the nearest city was regarded as a social covariate in a work about conflict risk modelling (Ref. 21).

- 1) Mach KJ, et al. Climate as a risk factor for armed conflict. *Nature* 571, 193-197 (2019).
- 2) Zhukov YM. Roads and the diffusion of insurgent violence: The logistics of conflict in Russia's North Caucasus. *Political Geography* 31, 144-156 (2012).

For other covariates, we edited them in a similar way as described above as well as the explanation of the data generation process for clarity. More modification details are to be found in

the revised data section (Pages 19-24, lines 378-491).

In addition, according to your third comment, “Also predictor variables selected may need rigorous statistical tests, such as collinearity test”, rigorous statistical tests (i.e., collinearity test) were adopted for the selected covariates. We did find some collinearity problems when we use the statistical tests. Thus, we made adjustments to the entire manuscript, as shown in the next response (Response to Reviewer’s comment No. 3).

Point No.3 Also predictor variables selected may need rigorous statistical tests, such as collinearity test.

Response to Reviewer’s comment No. 3: Thanks for your nice and valuable suggestion. According to your comment, rigorous statistical tests (i.e., collinearity test) were adopted for the selected covariates. Unfortunately, there are some collinearity problems between the drought index and several covariables (i.e., rainfall deviation and temperature deviation) with Variance Inflation Factor (VIF) ranges from 10 to 100. Thus, we adjusted some covariates. For the newly selected covariates, correlation matrix and VIF were calculated during the two modelling processes. In the first stage, one-year samples (2000) used in the first simulation process under four strategies are adopted to estimate correlation matrix and VIF, as shown in Table S2-S5. In the second stage, all samples used in the first simulation process under four strategies are adopted to estimate correlation matrix and VIF, as shown in Table S6-S9. Generally, these results illustrate that multicollinearity is unlikely to affect our analysis. The Statistical test section has been added in the Supplementary Information (Pages 5-8, Lines 63-118 in Supplementary Information). In addition, due to changes in the covariates, we re-analyzed the BRT modelling process and adjusted the results of the entire manuscript, as shown in the revised version (Pages 1-2, lines 20-37; Pages 7-24, lines 130-491).

Table S2. Correlation matrix between covariate variables used in BRT ensembles trained on one-year (2000) samples under strategy a.

	SPI	STI	E	NTL	UA	ED	NDH	MP	MT	NDVI
SPI	1	-0.304	-0.129	-0.053	0.087	-0.005	0.065	0.266	0.049	0.163
STI	-0.304	1	0.11	-0.059	-0.013	0.079	-0.053	-0.343	-0.031	-0.185
E	-0.129	0.11	1	-0.097	0.094	0.092	0.022	-0.135	-0.16	-0.169

NTL	-0.053	-0.059	-0.097	1	-0.171	0.107	0.008	0.007	0.105	0.046
UA	0.087	-0.013	0.094	-0.171	1	-0.295	-0.127	-0.184	-0.499	-0.36
ED	-0.005	0.079	0.092	0.107	-0.295	1	0.096	0.134	0.232	0.206
NDH	0.065	-0.053	0.022	0.008	-0.127	0.096	1	0.244	0.147	0.13
MP	0.266	-0.343	-0.135	0.007	-0.184	0.134	0.244	1	0.487	0.7
MT	0.049	-0.031	-0.16	0.105	-0.499	0.232	0.147	0.487	1	0.483
NDVI	0.163	-0.185	-0.169	0.046	-0.36	0.206	0.13	0.7	0.483	1

Note: SPI (Standardized precipitation index): VIF = 1.175; STI (Standardized temperature index): VIF = 1.260; E (Elevation): VIF = 1.091; NTL (Nighttime lights): VIF = 1.057; UA (Urban accessibility): VIF = 1.546; ED (Ethnic diversity): VIF = 1.161; NDH (Natural disaster hotspots) : VIF = 1.087; MP (Mean precipitation) : VIF = 2.607; MT (Mean temperature) : VIF = 1.768; NDVI (normalized difference vegetation index) : VIF = 2.273.

Table S3. Correlation matrix between covariate variables used in BRT ensembles trained on one-year (2000) samples under strategy a+.

	SPI	STI	E	NTL	UA	ED	NDH	MP	MT	NDVI
SPI	1	-0.384	-0.137	-0.059	0.043	0.006	0.084	0.363	0.16	0.223
STI	-0.384	1	0.149	0.002	-0.069	0.139	0.059	-0.333	-0.043	-0.232
E	-0.137	0.149	1	-0.097	0.094	0.092	0.022	-0.135	-0.16	-0.169
NTL	-0.059	0.002	-0.097	1	-0.171	0.107	0.008	0.007	0.105	0.046
UA	0.043	-0.069	0.094	-0.171	1	-0.295	-0.127	-0.184	-0.499	-0.36
ED	0.006	0.139	0.092	0.107	-0.295	1	0.096	0.134	0.232	0.206
NDH	0.084	0.059	0.022	0.008	-0.127	0.096	1	0.244	0.147	0.13
MP	0.363	-0.333	-0.135	0.007	-0.184	0.134	0.244	1	0.487	0.7
MT	0.16	-0.043	-0.16	0.105	-0.499	0.232	0.147	0.487	1	0.483
NDVI	0.223	-0.232	-0.169	0.046	-0.36	0.206	0.13	0.7	0.483	1

Note: SPI (Standardized precipitation index): VIF = 1.299; STI (Standardized temperature index): VIF = 1.342; E (Elevation): VIF = 1.092; NTL (Nighttime lights): VIF = 1.051; UA (Urban accessibility): VIF = 1.544; ED (Ethnic diversity): VIF = 1.176; NDH (Natural disaster hotspots): VIF = 1.107; MP (Mean precipitation): VIF = 2.567; MT (Mean temperature): VIF = 1.755; NDVI (normalized difference vegetation index): VIF = 2.275.

Table S4. Correlation matrix between covariate variables used in BRT ensembles trained on one-year (2000) samples under strategy b.

	SPI	STI	MP	MT
SPI	1	-0.304	0.266	0.049
STI	-0.304	1	-0.343	-0.031
MP	0.266	-0.343	1	0.487
MT	0.049	-0.031	0.487	1

Note: SPI (Standardized precipitation index): VIF = 1.143; STI (Standardized temperature index): VIF = 1.226;

MP (Mean precipitation): VIF = 1.579; MT (Mean temperature): VIF = 1.352.

Table S5. Correlation matrix between covariate variables used in BRT ensembles trained on one-year (2000) samples under strategy b+.

	SPI	STI	MP	MT
SPI	1	-0.384	0.363	0.16
STI	-0.384	1	-0.333	-0.043
MP	0.363	-0.333	1	0.487
MT	0.16	-0.043	0.487	1

Note: SPI (Standardized precipitation index): VIF = 1.266; STI (Standardized temperature index): VIF = 1.262;

MP (Mean precipitation): VIF = 1.581; MT (Mean temperature): VIF = 1.339.

Table S6. Correlation matrix between covariate variables used in BRT ensembles trained on all samples under strategy a.

	SPI	STI	E	NTL	UA	ED	NDH	MP	MT	NDVI
SPI	1	-0.106	0.022	-0.031	0.01	0.008	0.031	0.013	-0.03	0.026
STI	-0.106	1	0.01	-0.023	-0.037	-0.043	-0.005	-0.012	0.216	0.031
E	0.022	0.01	1	-0.108	0.06	0.111	0.002	-0.153	-0.155	-0.201
NTL	-0.031	-0.023	-0.108	1	-0.174	0.112	0.024	-0.009	0.11	0.027

UA	0.01	-0.037	0.06	-0.174	1	-0.285	-0.126	-0.138	-0.494	-0.261
ED	0.008	-0.043	0.111	0.112	-0.285	1	0.082	0.114	0.201	0.164
NDH	0.031	-0.005	0.002	0.024	-0.126	0.082	1	0.266	0.159	0.149
MP	0.013	-0.012	-0.153	-0.009	-0.138	0.114	0.266	1	0.419	0.736
MT	-0.03	0.216	-0.155	0.11	-0.494	0.201	0.159	0.419	1	0.372
NDVI	0.026	0.031	-0.201	0.027	-0.261	0.164	0.149	0.736	0.372	1

Note: SPI (Standardized precipitation index): VIF = 1.016; STI (Standardized temperature index): VIF = 1.098; E (Elevation): VIF = 1.099; NTL (Nighttime lights): VIF = 1.055; UA (Urban accessibility): VIF = 1.501; ED (Ethnic diversity): VIF = 1.147; NDH (Natural disaster hotspots) : VIF = 1.100; MP (Mean precipitation) : VIF = 2.591; MT (Mean temperature) : VIF = 1.732; NDVI (normalized difference vegetation index) : VIF = 2.399.

Table S7. Correlation matrix between covariate variables used in BRT ensembles trained on all samples under strategy a+.

	SPI	STI	E	NTL	UA	ED	NDH	MP	MT	NDVI
SPI	1	-0.093	0.017	-0.046	0.02	0.004	0.044	0.043	-0.022	0.051
STI	-0.093	1	0.022	-0.019	-0.038	-0.043	-0.007	-0.028	0.239	0.02
E	0.017	0.022	1	-0.108	0.06	0.111	0.002	-0.153	-0.155	-0.201
NTL	-0.046	-0.019	-0.108	1	-0.174	0.112	0.024	-0.009	0.11	0.027
UA	0.02	-0.038	0.06	-0.174	1	-0.285	-0.126	-0.138	-0.494	-0.261
ED	0.004	-0.043	0.111	0.112	-0.285	1	0.082	0.114	0.201	0.164
NDH	0.044	-0.007	0.002	0.024	-0.126	0.082	1	0.266	0.159	0.149
MP	0.043	-0.028	-0.153	-0.009	-0.138	0.114	0.266	1	0.419	0.736
MT	-0.022	0.239	-0.155	0.11	-0.494	0.201	0.159	0.419	1	0.372
NDVI	0.051	0.02	-0.201	0.027	-0.261	0.164	0.149	0.736	0.372	1

Note: SPI (Standardized precipitation index): VIF = 1.017; STI (Standardized temperature index): VIF = 1.122; E (Elevation): VIF = 1.101; NTL (Nighttime lights): VIF = 1.056; UA (Urban accessibility): VIF = 1.506; ED (Ethnic diversity): VIF = 1.147; NDH (Natural disaster hotspots): VIF = 1.100; MP (Mean precipitation): VIF = 2.607; MT (Mean temperature): VIF = 1.772; NDVI (normalized difference vegetation index): VIF = 2.402.

Table S8. Correlation matrix between covariate variables used in BRT ensembles trained on all samples under strategy b.

	SPI	STI	MP	MT
SPI	1	-0.106	0.013	-0.03
STI	-0.106	1	-0.012	0.216
MP	0.013	-0.012	1	0.419
MT	-0.03	0.216	0.419	1

Note: SPI (Standardized precipitation index): VIF = 1.012; STI (Standardized temperature index): VIF = 1.074; MP (Mean precipitation): VIF = 1.229; MT (Mean temperature): VIF = 1.289.

Table S9. Correlation matrix between covariate variables used in BRT ensembles trained on all samples under strategy b+.

	SPI	STI	MP	MT
SPI	1	-0.093	0.043	-0.022
STI	-0.093	1	-0.028	0.239
MP	0.043	-0.028	1	0.419
MT	-0.022	0.239	0.419	1

Note: SPI (Standardized precipitation index): VIF = 1.011; STI (Standardized temperature index): VIF = 1.091; MP (Mean precipitation): VIF = 1.241; MT (Mean temperature): VIF = 1.313.

Point No.4 Types of armed conflict events cannot elaborate leading factors in line with characteristics of natural conditions in target region.

Response to Reviewer’s comment No. 4: Thank you for your suggestion. In this study, the types of armed conflicts are not further distinguished. Data on armed conflicts are taken from the UCDP website, which is an openly available armed conflict dataset with georeferenced information. The maximum spatial resolution of the UCDP dataset is the individual village or town. Therefore, we can localize the armed conflicts to $0.1^{\circ} \times 0.1^{\circ}$ grids based on latitude and longitude coordinates.

Point No.5 In addition, the parameters of the Boosted Regression Tree model were not listed in

detail. Thus, supplements are highly recommended.

Response to Reviewer's comment No. 5: Thanks for your valuable suggestions. According to your suggestion, the parameters of the boosted regression tree model were listed in Supplementary Information, as follows:

“The following parameters are required to be determined during using BRT modelling framework: a) the complexity of individual trees (tree.complexity); b) the weight applied to individual trees (learning.rate); c) the proportion of observations used in selecting variables (bag.fraction); d) numbers of trees to add at each cycle (step.size); e) the number of folds cross-validation (cv.folds); f) max number of trees to fit before stopping (max.trees). For parameters a-f, we follow Bhatt et al. (2013) in setting tree.complexity equal to 4, learning.rate equal to 0.01, bag.fraction equal to 0.75, step.size equal to 10, cv.folds equal to 10, and max.trees equal to 10000. In addition, it should be noted that specifying the optimal number of trees plays an important role during the BRT modelling process. In the present study, the methods of Elith et al. (2008) was combined with 10-fold cross validation process to determine the optimal number of trees. Other parameters were held at their default values.” (Page 4, lines 38-49 in Supplementary Information).

- 1) Bhatt S, et al. The global distribution and burden of dengue. *Nature* 496, 504-507 (2013).
- 2) Elith J, Leathwick JR, Hastie T. A working guide to boosted regression trees. *Journal of Animal Ecology* 77, 802-813 (2008).

Point No.6 In the introduction, authors indicated that "exploring the underlying mechanisms is still a challenging task.", but no detail discussion can be seen in the section of Discussion.

Response to Reviewer's comment No. 6: Thank you for your valuable suggestion. Firstly, we rephrased the sentence by replacing“exploring the underlying mechanisms is still a challenging task” with “exploring the underlying mechanisms at the global scale is still a challenging task”. Secondly, we added several discussions about the underlying mechanisms in the section of discussion in the revised version, as follows:

“Our procedure allows for quantifying the relationship between covariates and armed conflicts at a global scale. Overall, the mined potential patterns derived from a large amount of data are complex, because different meteorologic, geographic, political and socioeconomic contexts may make human beings adapt differently to the environment (Refs. 29 and 39), leading to varying

social stability that responds differently to climate change. However, there are several universal patterns, as shown in Supplementary Figs. 4 and 5. For instance, the positive association between risk level and ethnic diversity illustrate that a greater diversity of the politically relevant ethnicity leads to higher risks of conflicts, which is consistent with several previous studies (Refs. 17, 31, 32 and 33). For the climate deviations related covariates, a few studies have suggested that negative temperature deviation in temperate locations may lead to conflict and negative precipitation deviation coinciding with social instability (Refs. 18, 34, 35 and 36). However, modern humans' adaptability to climate is much higher than that recorded in historical studies due to the increase in technological adaptability, and local regions (i.e., agriculturally dependent groups) or individual cases (i.e., Syrian civil war) cannot represent the contemporary relationship between climate change and conflict on a global scale. Fig. 2 and Supplementary Fig.6 indicate that that the positive temperature deviation or high precipitation extremes are associated with more conflict across the globe from 2000 to 2015, which vindicate the finding of a comprehensive quantitative study conducted by Hsiang et al. (Ref. 14). In addition, our findings reveal that temperature raising has a greater nonlinear impact on conflict risks than precipitation deviation at a global scale.” (Pages 15-16, lines 295-317)

Point No.7 English needs editing and proofreading.

Response to Reviewer's comment No. 7: According to your valuable comments, most statements have been modified. The revised manuscript has been spell-checked and grammar-checked by American Journal Experts (www.journalexperts.com) and William Jefferson.

Reviewer #3 (Remarks to the Author):

The paper proposes a machine learning-based approach to simulate the global armed conflict risk at $0.1^{\circ} \times 0.1^{\circ}$ spatial resolution for the period from 2000-2015.

The paper must be improved in the following aspects:

Point No.1 The literature review needs to be updated with more papers published in this three last years, since machine learning is a hot research topic

Response to Reviewer's comment No. 1: Thanks for your valuable suggestions. According to

your comments, the literature reviewed in the introduction section of this manuscript have been updated using four papers published in the period from 2019 to 2020 (Pages 5-6, lines 111-116).

The updated literature are as follows:

- 1) Hegre H, et al. ViEWS: A political violence early-warning system. *Journal of Peace Research* 56, 155-174 (2019).
- 2) Reichstein M, et al. Deep learning and process understanding for data-driven Earth system science. *Nature* 566, 195-204 (2019).
- 3) Yang KK, Wu Z, Arnold FH. Machine-learning-guided directed evolution for protein engineering. *Nature Methods* 16, 687-694 (2019).
- 4) Jiang Z, et al. Machine-learning-revealed statistics of the particle-carbon/binder detachment in lithium-ion battery cathodes. *Nature Communications* 11, 2310 (2020).

Point No. 2 The authors must provide a flowchart describing the used methodology

Response to Reviewer's comment No. 2: Thanks for your nice suggestions. In the revised supplementary Information, an analytic process overview was added, as shown in Supplementary Fig. 1 (Page 10, lines 140-145 in Supplementary Information). The process used to simulate armed conflict risk at a global scale involved two stages. In the first stage (black arrow), the input dataset was combined with BRT modelling framework to prove the hypothesis. If it was proved to be true, the second stage (blue arrow) would start the analysis. Otherwise, the analytic process would end. In addition, the sentence, "The analytic process overview is shown in Supplementary Fig. 1", is added in the methods section in the revised version (Page 19, lines 365-366).

Point No. 3 The Table 1 (describing the relative importance of covariates in predicting the global risk of armed conflict events based on the 20 ensemble BRT models trained on all samples from period 2000-2015) must be enriched and verified with knowledge domains from experts.

Response to Reviewer's comment No. 3: Thank you for your valuable comments. In the revised version, Fig.2 (Page 9, lines 178-183) and Supplementary Figs. 4-6 (Pages 12-13, lines 155-175 in Supplementary Information) are added to replace Table 1. These sub-plots are ordered by the mean relative contribution (%) of covariates, with these mean relative contribution \pm standard deviation (%) given within each sub-plot. Meantime, these sub-plots depict the marginal effect

curves of each covariate over the BRT ensembles fitted to the full samples under strategies a and a+. In addition, Table 1 used in the old version is put into Supplementary Information is also renamed as Supplementary Table 4 (Page 19, lines 236-239). In the revised version, related content has been enriched and verified with experts' knowledge and previous literature in the discussion section (Pages 14-16, lines 269-317). The details are as follows:

“Previous studies conducted by O’Loughlin et al. suggested that although climate anomalies are associated with conflict risk, the conflict risk is influenced more by political, socioeconomic, and geographic contexts than by climate anomalies, especially in sub-Saharan Africa (Refs. 15 and 25). Our findings reveal that there are similar patterns at the global scale. For instance, the stable background covariates greatly contributed to the spatial-temporal distribution of armed conflicts with a mean relative contribution of more than 96.000%. Compared with the stable background covariates, standardized temperature index or standardized precipitation index had relatively little effect on the simulated results, but climate deviation related covariates cumulatively accounted for more than 2.500% of the simulated results. This is the reason that the simulated risk level of armed conflict in the local regions varies in different years. We interpret this as evidence of the connection between conflict events and climate change. Supplementary Table 2 suggests that considering the 24-month climate deviations can slightly improve the performance of the BRT models. This result can be seen as partially supporting other research findings (Refs. 26 and 27) that sequential multi-year climate deviations from normal may in-part or by-whole affect the collapse of civilizations. In addition, Supplementary Table 4 reveals that the long time period climate deviations have a greater impact on risk level with the relative contribution values of 3.806%, which is in line with the diverse disciplines experts’ judgments that 3–20% of conflict risk links to climate variability (Ref. 28). (Pages 14-15, lines 269-294)

Our procedure allows for quantifying the relationship between covariates and armed conflicts at a global scale. Overall, the mined potential patterns derived from a large amount of data are complex, because different meteorologic, geographic, political and socioeconomic contexts may make human beings adapt differently to the environment (Refs. 29 and 39), leading to varying social stability that responds differently to climate change. However, there are several universal patterns, as shown in Supplementary Figs. 4 and 5. For instance, the positive association between risk level and ethnic diversity illustrate that a greater diversity of the politically relevant ethnicity

leads to higher risks of conflicts, which is consistent with several previous studies (Refs. 17, 31, 32 and 33). For the climate deviations related covariates, a few studies have suggested that negative temperature deviation in temperate locations may lead to conflict and negative precipitation deviation coinciding with social instability (Refs. 18, 34, 35 and 36). However, modern humans' adaptability to climate is much higher than that recorded in historical studies due to the increase in technological adaptability, and local regions (i.e., agriculturally dependent groups) or individual cases (i.e., Syrian civil war) cannot represent the contemporary relationship between climate change and conflict on a global scale. Fig. 2 and Supplementary Fig.6 indicate that that the positive temperature deviation or high precipitation extremes are associated with more conflict across the globe from 2000 to 2015, which vindicate the finding of a comprehensive quantitative study conducted by Hsiang et al. (Ref. 14). In addition, our findings reveal that temperature raising has a greater nonlinear impact on conflict risks than precipitation deviation at a global scale.” (Pages 15-16, lines 295-317)

REVIEWER COMMENTS

Reviewer #2 (Remarks to the Author):

Authors answered what I concerned to, and the readability of the revised manuscript is improved a lot. It is suggested to accept it for publication.

Reviewer #3 (Remarks to the Author):

The submitted paper may be accepted in this revised version according to the comments of other reviewers.

However, a minor comment concerns the need to provide more interpretability and explainability of the used machine learning methods.

Reviewer #4 (Remarks to the Author):

This is a technically well-executed paper, adding to a growing literature on conflict prediction and climate in the field. I have three main comments.

[1] Novelty and importance of the contribution

The biggest strength of this paper is that the authors may be the first to run this subnational-level prediction exercise at the global level with climate and socio-economic data. There have been similar exercises focusing on the causal identification of climate fluctuations on conflict risk. And there have been several subnational exercises to predict conflict, often with a wider array of predictors. But I am not aware of a similar exercise to the authors. I do not keep up with 100% of this literature, so I am not certain this is the first paper of this nature. But if they are correct, then this is a useful new analysis.

At the same time, there are some weaknesses to this contribution.

First, my personal feeling is that prediction exercises have yet to really make a substantive contribution to our understanding of conflict. It's not clear what the theory is that connects climate fluctuations to an increased number and intensity of battles, and it's not clear what light a predictive analysis sheds on this.

I think there is a practical use for prediction exercises, which is to help test and develop the potential for early warning systems. But I think, to be effective, these prediction exercises / early warning systems have to draw on a wider array of covariates. This brings us to the second weakness, which is that a study focusing mainly on climate variables alone is of more limited usefulness in this early warning exercise. I think it is why most of the other studies have focused on a single country or region with richer predictors.

Thus, despite the novelty and impressiveness of the analysis, I have trouble articulating what we learn that is new. The authors find that the largest predictors of conflict events are relatively time-invariant, and that climatic shocks have more modest explanatory or predictive power. This seems to be consistent with what the large conflict and climate literature has established in the past. Indeed, the finding is consistent with a broader literature on predicting conflict that has struggled to find time-varying predictors of conflict risk (e.g. work by Samuel Bazzi, Robert Blair, and coauthors). And it is consistent with a large causal literature that finds small but significant effects of climate shocks on conflict (as in the Burke, Hsiang and Miguel review article).

I think the authors need to make a much clearer and more specific case about the substantive, practical, or theoretical contribution of their analysis.

[2] What is being forecasted?

I think the authors could be more specific in what they are predicting. There is a significant theoretical and practical difference between the outbreak of hostilities (a broader, long lasting conflict) and the escalation or intensification of hostilities (increasing size and frequency of conflict events). In short, the reasons two groups go to war are often quite different than the reasons that battles and skirmishes happen, and thus they probably have distinct drivers. My reading of the paper and the data source is that most of the events forecasted are largely in ongoing conflicts. Thus, it is probably not quite right to say in the abstract that the authors “simulate the risk of armed 23 conflicts worldwide from 2000-2015” or that “Results revealed that the conflict risks are primarily influenced by...” (p1). I think the authors may want to pay attention to the nuance here. I think it's important to say they they simulate the risk of conflict *events*.

[3] Describe the data generating process and potential measurement error/missingness

Many conflict papers suffer from the problem of not knowing the data. There's no description of the data sources or reliability. How do UCDP get these data? What are the strengths and weaknesses? I am not intimately familiar with the georeferenced datasets by UCDP or ACLED, but a few cautions about these georeferenced datasets are in order:

- They typically rely mainly on press accounts in countries with very limited press, and often no journalists in the areas with fighting
- In most countries it seems they use only the English language press, and perhaps international press only. It's unclear.
- Many newspapers are state owner or at least nonpartisan, meaning reports battles can be skewed
- When a story does appear in the press, my understanding is that these datasets omit any event that does not have a specific location

Altogether, this suggests to me that the vast majority of events are probably omitted from these datasets. These are unlikely to be missing at random. Thus any paper that uses them has to think about the missingness and speculate how the statistical analysis could be affected by such missingness. It's plausible that missingness is not related to climatic variability, but this deserves to be discussed. A clear and candid discussion will enhance the credibility of the paper rather than detract from it.

Responses to reviewers

NCOMMS-20-07482A

Title: Modelling armed conflict events with machine learning and high-frequency time-series data

We sincerely thank all the reviewers for their critical reading and constructive comments and suggestions for improving our manuscript. We have carefully addressed all the comments and questions. Answers are typed in **blue**. Following the reviewers' comments and suggestions, we have revised the manuscript accordingly, as detailed below.

Reviewer #2 (Remarks to the Author):

Authors answered what I concerned to, and the readability of the revised manuscript is improved a lot. It is suggested to accept it for publication.

Response to Reviewer's comment: Thank you for your helpful comments and constructive suggestions for improving our manuscript.

Reviewer #3 (Remarks to the Author):

The submitted paper may be accepted in this revised version according to the comments of other reviewers. However, a minor comment concerns the need to provide more interpretability and explainability of the used machine learning methods.

Response to Reviewer's comment: Thank you for your helpful suggestions for improving our manuscript. We appreciate your attention to the need for more interpretability and explainability of the used machine learning methods. We agreed that exploring the links between covariates and armed conflict events could enhance the significance of the manuscript. Therefore, we followed your suggestions and modified the related statements to discuss the mined links and to explain the reasons for differences compared with previous studies, as follows:

“Our procedure allows for quantifying the relationship between covariates and armed conflict events at a global scale. Overall, the mined potential patterns derived from a large amount of data are complex, since different meteorologic, geographic, political and socioeconomic contexts may make human beings adapt differently to the environment (Refs. 29 and 30), leading to varying social stability that responds differently to climate change. However, there are several universal

patterns, as shown in Supplementary Figs. 4 and 5. For instance, the positive association between risk level and ethnic diversity illustrate that a greater diversity of the politically relevant ethnicity leads to higher risks of conflict events, which is consistent with several previous studies (Refs. 17, 31, 32 and 33). For the climate deviations related covariates, a few studies have suggested that negative temperature deviation in temperate locations may lead to conflict and negative precipitation deviation coinciding with social instability (Refs. 18, 35, 36 and 37). However, modern humans' adaptability to climate is much higher than that recorded in historical studies due to the improvement in technological adaptability and the increase in complexity of social structure, whereas local regions (i.e., agriculturally dependent groups) or individual cases (i.e., Syrian civil war) cannot represent the contemporary relationship between climate change and conflict on a global scale. Fig. 2 and Supplementary Fig.6 indicate that that the positive temperature deviation or high precipitation extremes are associated with more conflict across the globe from 2000 to 2015, which vindicate the finding of a comprehensive quantitative study conducted by Hsiang et al. (Ref.14)” (Pages 11-12, Lines 220-242)

Also, we further provided more interpretability of the models in the revised discussion section. The details are as follows:

“Meanwhile, there is a negative link between risk level and urban accessibility, revealing that transportation hubs can easily become the outbreak site of conflict since they play a key role in control territories (Ref. 34).” (Pages 11-12, lines 229-231)

Reviewer #4 (Remarks to the Author):

This is a technically well-executed paper, adding to a growing literature on conflict prediction and climate in the field. I have three main comments.

Point No.1 [1] Novelty and importance of the contribution

The biggest strength of this paper is that the authors may be the first to run this subnational-level prediction exercise at the global level with climate and socio-economic data. There have been similar exercises focusing on the causal identification of climate fluctuations on conflict risk. And there have been several subnational exercises to predict conflict, often with a wider array of predictors. But I am not aware of a similar exercise to the authors. I do not keep up with 100% of

this literature, so I am not certain this is the first paper of this nature. But if they are correct, then this is a useful new analysis.

At the same time, there are some weaknesses to this contribution.

First, my personal feeling is that prediction exercises have yet to really make a substantive contribution to our understanding of conflict. It's not clear what the theory is that connects climate fluctuations to an increased number and intensity of battles, and it's not clear what light a predictive analysis sheds on this.

I think there is a practical use for prediction exercises, which is to help test and develop the potential for early warning systems. But I think, to be effective, these prediction exercises / early warning systems have to draw on a wider array of covariates. This brings us to the second weakness, which is that a study focusing mainly on climate variables alone is of more limited usefulness in this early warning exercise. I think it is why most of the other studies have focused on a single country or region with richer predictors.

Thus, despite the novelty and impressiveness of the analysis, I have trouble articulating what we learn that is new. The authors find that the largest predictors of conflict events are relatively time-invariant, and that climatic shocks have more modest explanatory or predictive power. This seems to be consistent with what the large conflict and climate literature has established in the past. Indeed, the finding is consistent with a broader literature on predicting conflict that has struggled to find time-varying predictors of conflict risk (e.g. work by Samuel Bazzi, Robert Blair, and coauthors). And it is consistent with a large causal literature that finds small but significant effects of climate shocks on conflict (as in the Burke, Hsiang and Miguel review article).

I think the authors need to make a much clearer and more specific case about the substantive, practical, or theoretical contribution of their analysis.

Response to Reviewer's comment No. 1: We are very grateful for your insightful suggestions. Our research is the first to use climate and socio-economic data to make such sub-national predictions at the global level. An advanced machine learning-based modelling framework was set up to reveal the potential links between covariates and conflict events on a global scale. The method performed well with high accuracy and stability. In addition, the contributions of the key factors were quantitatively described and discussed. In the revised version, we followed your suggestions and added several statements to further indicate the novelty and importance of the

contribution. The details are as follows:

“Our results reveal that the risk of conflict events are primarily influenced by stable background contexts with complex patterns, followed by climate deviations related covariates. The inferred patterns show that positive temperature deviation or high precipitation extremes are associated with increased risk of conflict events worldwide. Our findings indicate that a better understanding of climate-conflict link at the global scale enhances the spatiotemporal modelling capacity for the risk of armed conflict events.” (Page 2, Lines 28-38)

“In the aggregate, our study provides a better understanding of a climate–conflict link at the global scale and enhances the spatial-temporal modelling capacity for the risk of armed conflict events worldwide.” (Pages 13-14, Lines 273-275)

As you mentioned, focusing on a single country or region and using more richer predictors (i.e., cultural and historical factors) are useful for prediction exercises. We did use many variables initially, but this raised two concerns during prediction exercises. On one hand, despite the abundance of the predictors, the collinearity among the predictors may impact the reliability of the analysis. On the other hand, the spatial resolution of several datasets (i.e., government finance) was low, which reduced the spatial resolution of the prediction results. Therefore, we finally combined the collinearity test and the availability of global fine-scale data to filter covariates. Meanwhile, analysis on the global scale brought more samples, which could give machine learning models more variation to train on and help machine learning models identify more reliable relationships. Therefore, global-scale modelling analysis with carefully selected predictors could also generate reasonable and reliable results. This could be proved by a comparative analysis. To further verify the early warning capabilities, we made predictions in Africa (Supplementary Fig. 15) and compared the predicted risk level with that of the previous study. The results illustrate that the predicted risks of conflict events in Africa (Supplementary Fig. 15b) are generally consistent with the risk level estimated by Hegre et al. (2019) (Ref. 21). In the revised version, several statements were added in the discussion section., as follows:

“Second, we have based our analysis on the global-scale multi-dimensional spatial-temporal refined dataset. Due to the availability of refined dataset of cultural and historical factors, the covariates adopted in this study are not the most comprehensive, however, with more samples for machine learning models to train on, our global-scale refined analysis can help models capture

more reliable relationships.”(Page 13, Lines 264-268)

“Our modelling framework may be helpful for early warning of conflict events. This is indicated by the comparative analysis that the predicted risks of conflict events in Africa (Supplementary Fig. 15b) are generally consistent with the risk level estimated by Hegre et al. (Ref. 21)”(Page 13, Lines 269-273)

[1] Hegre H, Allansson M, Basedau M, et al. ViEWS: a political violence early-warning system[J]. Journal of peace research, 2019, 56(2): 155-174.

We agree with you that a theoretical explanation of why and how climate affects human conflicts across different contexts is of great importance. According to the widely agreed definition, a general theory is about to reveal the causality. Causality usually consists of two interrelated parts: (a) causal effects; and (b) causal mechanisms. Quantitative or semi-quantitative experiments help reveal causal effect but give little indication of causal mechanism. While the single-case methods aim to gauge causal mechanism, they say little about causal effect. In a word, case study (qualitatively) uses process-tracing method to reveal causal mechanism, while large-N studies (quantitatively) reveal causal effects. Currently, in social science, with an increasing number of various kinds of data, quantitative studies with the aim to reveal causal effects are preferred. Our research is the first research to quantify the causal effect between climate-conflict link across different contexts on a global scale. There were previous single-case studies showing that climatic changes may influence conflict through both economic and non-economic pathways, including possible psychological channels, but these causal mechanisms were derived from local regions or single-case, which were unable to explain why and how climate affects human conflict on the global scale. Based on your constructive suggestions, we added a caveat in the revised version. The details are as follows:

“Third, there is no general theory to explain the causal mechanism of the climate–conflict link at the global scale.”(Page 13, Lines 268-269)

Point No.2 [2] What is being forecasted?

I think the authors could be more specific in what they are predicting. There is a significant theoretical and practical difference between the outbreak of hostilities (a broader, long lasting conflict) and the escalation or intensification of hostilities (increasing size and frequency of

conflict events). In short, the reasons two groups go to war are often quite different than the reasons that battles and skirmishes happen, and thus they probably have distinct drivers. My reading of the paper and the data source is that most of the events forecasted are largely in ongoing conflicts. Thus, it is probably not quite right to say in the abstract that the authors “simulate the risk of armed conflicts worldwide from 2000-2015” or that “Results revealed that the conflict risks are primarily influenced by...” (p1). I think the authors may want to pay attention to the nuance here. I think it’s important to say they they simulate the risk of conflict **events**.

Response to Reviewer’s comment No. 2: We appreciate your attention to the nuance between “conflict” and “conflict events”. Based on your valuable comments, we have reviewed the conflict events dataset and recheck the relevant statements in the manuscript. In the revised version, we followed your suggestions and made a corresponding revision in the abstract section, as follows:

“Here we adopt a formal machine learning-based modelling framework to infer potential mechanisms from high-frequency time-series data and simulate the risk of conflict events worldwide from 2000-2015. Our results reveal that the risk of conflict events are primarily influenced by stable background contexts with complex patterns, followed by climate deviations related covariates. The inferred patterns show that positive temperature deviation or high precipitation extremes are associated with increased risk of conflict events worldwide. Our findings indicate that a better understanding of climate-conflict link at the global scale enhances the spatiotemporal modelling capacity for the risk of armed conflict events.” (Pages 1-2, Lines 25-38)

In addition, we also rephrase other related statements accordingly in Supplementary Information and the rest of the manuscript, including the revised introduction, results, and discussion sections.

Point No.3 [3] Describe the data generating process and potential measurement error/missingness
Many conflict papers suffer from the problem of not knowing the data. There’s no description of the data sources or reliability. How do UCDP get these data? What are the strengths and weaknesses? I am not intimately familiar with the georeferenced datasets by UCDP or ACLED, but a few cautions about these georeferenced datasets are in order:

- They typically rely mainly on press accounts in countries with very limited press, and often no journalists in the areas with fighting
- In most countries it seems they use only the English language press, and perhaps international press only. It's unclear.
- Many newspapers are state owner or at least nonpartisan, meaning reports battles can be skewed
- When a story does appear in the press, my understanding is that these datasets omit any event that does not have a specific location

Altogether, this suggests to me that the vast majority of events are probably omitted from these datasets. These are unlikely to be missing at random. Thus any paper that uses them has to think about the missingness and speculate how the statistical analysis could be affected by such missingness. It's plausible that missingness is not related to climatic variability, but this deserves to be discussed. A clear and candid discussion will enhance the credibility of the paper rather than detract from it.

Response to Reviewer's comment No. 3: Thanks for your valuable suggestions. UCDP GED and ACLED are two leading conflict events datasets. We had a preliminary understanding of these two types of data before this study. The quality of UCDP GED's geocoding and precision information is far superior to ACLED's, which is particularly important for us to analyze geographic dimensions of armed conflict. Therefore, UCDP GED was adopted in the present study. However, UCDP GED is sometimes criticized for its reliance on media sources, as you mentioned. We have also checked the quality of UCDP GED carefully before. In order to alleviate the well-known media bias, UCDP GED does not rely solely on media reports, but also on NGO reports, case studies, databases and historical archives, as described in previous literature (Ralph Sundberg and Erik Melander, 2013; Kristine Eck, 2012). In addition, triple-checked was employed to improve the quality of the final dataset (Ralph Sundberg and Erik Melander, 2013; Kristine Eck, 2012). For example, the first manual check is done by the coder, the second by the UCDP project leader, and the third check is done by automated scripts.

It is also important to note that UCDP GED cannot resolve the bias completely and include all armed conflict events. In the revised version, we followed your nice suggestions and added a clear and candid discussion about the concern to enhance the credibility of the manuscript. The details are as follows:

“In this study, there are some caveats. First, media reports represent one source of data for UCDP GED, and well-known media bias may add uncertainty to our results at a certain extent. Although several measures (i.e., triple-checked) were employed to ensure high quality of final dataset (Ref. 38), UCDP GED cannot resolve the bias completely and include all armed conflict events.” (Page 13, Lines 259-263)

[1] Sundberg R, Melander E. Introducing the UCDP georeferenced event dataset[J]. Journal of Peace Research, 2013, 50(4): 523-532.

[2] Eck K. In data we trust? A comparison of UCDP GED and ACLED conflict events datasets[J]. Cooperation & Conflict, 2012, 47(1):124-141.

REVIEWER COMMENTS

Reviewer #4 (Remarks to the Author):

I don't think the authors really addressed my suggestions. I explain a bit more below. I leave it to the discretion of the editor which are required.

To me the critical change needed is for the authors to be precise with their terms and to be clear about what is being predicted.

I see three sources of imprecision:

First, nowhere in the main paper do the authors define a their dependent variable, conflict events.

Second, elsewhere in the paper, the authors use the term "conflict risk". In political science and economics this term generally refers to the risk of a peaceful competition between two rivals runs into violent competition—conflict onset, in formal terms. The authors are not forecasting outbreak. Most likely, 99.9% of the conflict events in the dataset are events that happen after the initial outbreak of war. Hence, the authors are predicting, conditional on a state of war, what predicts events. This is an important distinction, because there is also an important and conceptually distinct literature predicting the onset of new conflicts.

Third, the discussion of the literature doesn't distinguish between papers predicting conflict onset, conflict events, or other dependent variables.

I think three simple clarifications will fix this.

First, I recommend defining conflict events within the first page.

Second, I suggest that the introduction or discussion emphasize clearly that the paper is not predicting new conflict onset, but rather predicting events conditional on a state of war.

Third, when a paper is discusses, be clear if it is predicting conflict events, onsets, or some other variable. Well-established terms exist in the literature to help make these distinctions.

Finally, I also made the point in my last review that the data should be explained. In economics and political science, it would be customary in a short form paper to discuss the data—perhaps a 1/2 page to a page—in the supplementary materials.

Responses to reviewers

NCOMMS-20-07482B

Title: Modelling armed conflict events with machine learning and high-frequency time-series data

We sincerely thank all the reviewers for their critical reading and constructive comments and suggestions for improving our manuscript. We have carefully addressed all the questions mentioned in comments. In this revised version, we further clarified the terms and the dependent variable. Also, inspired by the reviewers, we carried out a supplementary modeling analysis for armed conflict onset. Answers are typed in **blue**. Following the reviewers' comments and suggestions, we have revised the manuscript accordingly, as detailed below.

Reviewer #4 (Remarks to the Author):

I don't think the authors really addressed my suggestions. I explain a bit more below. I leave it to the discretion of the editor which are required.

To me the critical change needed is for the authors to be precise with their terms and to be clear about what is being predicted.

I see three sources of imprecision:

First, nowhere in the main paper do the authors define a their dependent variable, conflict events.

Second, elsewhere in the paper, the authors use the term "conflict risk". In political science and economics this term generally refers to the risk of a peaceful competition between two rivals runs into violent competition—conflict onset, in formal terms. The authors are not forecasting outbreak. Most likely, 99.9% of the conflict events in the dataset are events that happen after the initial outbreak of war. Hence, the authors are predicting, conditional on a state of war, what predicts events. This is an important distinction, because there is also an important and conceptually distinct literature predicting the onset of new conflicts.

Third, the discussion of the literature doesn't distinguish between papers predicting conflict onset, conflict events, or other dependent variables.

I think three simple clarifications will fix this.

Point No.1 First, I recommend defining conflict events within the first page.

Response to Reviewer's comment No. 1: We are very grateful to you for pointing out the

inaccuracies and providing suggestions to improve our manuscript. According to your valuable advice, we rechecked the corresponding expression and found several statements that were unclearly described. We fully agree with you that defining conflict events can help to express the dependent variable more accurately. Thus, we distinguished the armed conflict incidence from the armed conflict onset in the revised introduction section, and as stated in our manuscript, we mainly focuses on the former. The details are as follows:

“According to the widely agreed definition, armed conflict risks involve armed conflict incidence and armed conflict onset, and this study mainly focuses on the former.” (Pages 2-3, Lines 47-49)

“c.) quantify the causal effect between climate-conflict link and simulate the likelihood of armed conflict incidence at the global scale. d.) combine the machine learning models with high-frequency time-series data to explore the feasibility of simulating armed conflict onset.”

(Page 5, Lines 110-114)

Meantime, we rephrased the corresponding statements throughout this revised manuscript to make a more precise definition of this term and clarify the dependent variable. The details are as follows:

“Based on UCDP GED, two binary dependent variables, including armed conflict incidence and armed conflict onset, were defined for each $0.1^\circ \times 0.1^\circ$ grid on a yearly basis to represent armed conflict risk. If there are one or more instances of armed conflict event in one grid in a single year, the armed conflict incidence indicator is coded as one (high-risk) for the grid. In addition, if a new armed conflict event outbreak after one calendar year of inactivity in one grid, armed conflict onset is assigned the value of one for the grid. Both binary dependent variables are otherwise assigned the value of zero (low-risk). For each year, an equivalent amount of low-risk samples and high-risk samples are randomly selected to construct the one-year samples and to train the BRT models. The detailed description of BRT modelling framework, independent variables and dependent variables can be found elsewhere (Supplementary Information).” (Page 15, Lines 309-320)

In addition, the dependent variable section was added in Supplementary Information to define armed conflict event and further clarify the dependent variable. The details are shown in Response to Reviewer’s comment No. 4.

Point No.2 Second, I suggest that the introduction or discussion emphasize clearly that the paper is not predicting new conflict onset, but rather predicting events conditional on a state of war.

Response to Reviewer’s comment No. 2: Thanks for your valuable suggestions. As mentioned above, in the revised introduction section, we declared that this manuscript mainly focused on simulating the likelihood of armed conflict incidence. Also, in the revised discussion section, we put in some discussions about armed conflict incidence and armed conflict onset. The details are shown in Response to Reviewer’s comment No. 3.

Point No.3 Third, when a paper is discusses, be clear if it is predicting conflict events, onsets, or some other variable. Well-established terms exist in the literature to help make these distinctions.

Response to Reviewer’s comment No. 3: Thanks for your nice and valuable comments. According to your suggestions, we made a careful modification, as following:

Firstly, based on the well-established terms existing in the previous literature, we made a distinction between armed conflict incidence and armed conflict onset in the revised version. According to the equations [1] and [2] adopted by Van Weezel, it is not hard to find that there are some differences between armed conflict incidence and armed conflict onset.

$$\text{Armed conflict incidence} = \begin{cases} 1 & \text{if armed conflict event in year } t \\ 0 & \text{if no armed conflict event in year } t \end{cases} \quad [1]$$

$$\text{Armed conflict onset} = \begin{cases} 1 & \text{if armed conflict event in year } t \text{ but not in year } t - 1 \\ 0 & \text{if no armed conflict event in year } t \text{ and year } t - 1 \end{cases} \quad [2]$$

- 1) Van Weezel, S. Economic shocks & civil conflict onset in Sub-Saharan Africa, 1981–2010. Defence and Peace Economics 26, 153-177 (2015).

Secondly, inspired by your suggestion, a supplementary modeling analysis for the dependent variable armed conflict onset was also conducted. The machine learning models were combined with high-frequency time-series data to infer the causal effect of climate on the conflict, and to simulate the risk of armed conflict onset (Pages 9-34, lines 138-447 in Supplementary Information). The analytic process of armed conflict onset is shown in Tables S6-S9 and S14-S17, Supplementary Figs 4, 5, 9-12, 17-20 and 25-28 and Supplementary Tables 2, 4, 6, and 8. The results showed that combining machine learning with high-frequency time-series data has great potential in predicting the risk of armed conflict onset. These contents were added in the revised

discussion section, as following:

“Several quantitative studies point out that armed conflict risks need to be further refined (Supplementary Refs. 18, 29 and 30). Our study not only simulates the likelihood of armed conflict incidence, but also further explores the feasibility of simulating armed conflict onset. Based on the definitions adopted by Stijn et al. (Supplementary Ref. 31), we constructed an incidence and an onset indicator to represent armed conflict risks and carried out modeling analysis separately. The findings further suggest that combining machine learning with high-frequency time-series data has great potential in predicting the risk of armed conflict onset at a global scale (Supplementary Figs. 4, 17 and 18). In addition, our results also indicate that armed conflict onset is more vulnerable to climate change than armed conflict incidence at a global scale, as shown in Supplementary Tables 7 and 8.” (Pages 11-12, Lines 229-239).

Thirdly, we rephrased several statements in the revised Results and Discussion sections, as following:

“Fig. 1 Validation performance on a time scale of the boosted regression trees (BRT) models trained on one-year incidence samples under strategies a (a) and a+ (b).” (Page 6, Lines 117-118)

“Based on UCDP GED and high-frequency time-series covariate dataset, we constructed a series of armed conflict incidence samples and armed conflict onset samples under the four strategies” (Page 6, Lines 122-124)

“Based on the 20 simulation processes, pairing stable background contexts with climate variability can simulate the spatial-temporal dynamics of armed conflict incidence well, as evidenced by the 10-fold cross-validation ROC-AUC (0.937±0.001 s.e.)” (Page 7, Lines 140-143)

“The performance assessment of the 20 ensemble BRT models trained on all samples under four strategies are described in Supplementary Tables 3 and 4. Supplementary Tables 5 and 6 illustrated that the significant differences (* means $p < 0.05$) that were observed for the performance of the BRT models trained on all samples under strategy a+ was comparable to those of strategy a” (Page 7, Lines 145-150)

“The relative contributions of each covariate for the 20 ensemble BRT models trained on all onset samples under strategies a and a+ are shown in Supplementary Figs. 9-12.” (Page 9, Lines 174-176)

“The final risk level maps derived from the mean of 20 ensemble BRT models trained on all incidence samples or all onset samples are shown in Supplementary Figs. 13-20, respectively.”

(Page 10, Lines 203-206)

“In addition, our findings reveal that rising temperature has a greater nonlinear impact on the risks of armed conflict incidence and armed conflict onset than precipitation deviation at a global scale.

Based on high-dimensional datasets and large volumes of occurrence records, we used the BRT models to simulate the global risks of armed conflict incidence and armed conflict onset at a grid-year level ($0.1^\circ \times 0.1^\circ$) under four strategies.” (Page 13, Lines 263-269)

Point No.4 Finally, I also made the point in my last review that the data should be explained. In economics and political science, it would be customary in a short form paper to discuss the data—perhaps a 1/2 page to a page—in the supplementary materials.

Response to Reviewer’s comment No. 4: Thanks for your nice and valuable comments. According to your suggestions, we further added Dependent Variable and Independent Variable sections in the revised supplementary materials to describe the data generating process as well as explaining why we chose these datasets, as following:

“**Dependent Variable:** Data on armed conflicts are taken from the openly available Uppsala Conflict Data Program (UCDP) georeferenced event dataset (GED). The UCDP GED is an armed conflict event dataset that includes state-based conflict, non-state conflict, and one-sided violence, and each armed conflict event is defined as: ‘The incidence of the use of armed force by an organized actor against another organized actor, or against civilians, resulting in at least 1 direct death at a specific location and for a specific temporal duration’ (Supplementary Ref. 1). In order to alleviate the well-known media bias, UCDP GED does not rely solely on media reports, but also on NGO reports, case studies, databases and historical archives. In addition, triple-checked was employed to improve the quality of the final dataset. In contrast to most other event datasets, the quality of UCDP GED’s geocoding and precision information is much better (Supplementary Ref. 2), which is particularly important for us to analyze geographic dimensions of armed conflict. Therefore, UCDP GED was adopted in the present study. Based on UCDP GED, we aggregate armed conflict events to the grid-year level and code two binary dependent variables (armed

conflict incidence and armed conflict onset) to represent the risk of armed conflicts. The two indicators are coded using the following equations [1] and [2] (Supplementary Ref. 3):” (Pages 3, lines 19-35 in Supplementary Information).

$$\text{Armed conflict incidence} = \begin{cases} 1 & \text{if armed conflict event in year } t \\ 0 & \text{if no armed conflict event in year } t \end{cases} \quad [1]$$

$$\text{Armed conflict onset} = \begin{cases} 1 & \text{if armed conflict event in year } t \text{ but not in year } t - 1 \\ 0 & \text{if no armed conflict event in year } t \text{ and year } t - 1 \end{cases} \quad [2]$$

“**Independent Variable:** Previous studies have linked armed conflicts to a series of covariates (Supplementary Refs. 4 and 5). For instance, politically relevant ethnic diversity might play a prominent role in conflict-prone regions, particularly in Africa and Asia, thus serving as a predetermined conflict line (Supplementary Ref. 6). In addition, climate change could worsen instability in volatile regions, especially in Africa (Supplementary Refs. 7-9). In the past decades, some interdisciplinary groups of scientists adopted various covariates to understand armed conflicts and predict the risk of armed conflicts; these primarily focused on a single country or region scale (Supplementary Refs. 10-12). However, the grid-year level ($0.1^\circ \times 0.1^\circ$) prediction exercise at the global scale remains a huge challenge due to the complexity of the mechanism and the availability of high-quality data. With an increasing number of various kinds of data and the further development of machine learning approaches, quantifying the causal effect between climate-conflict link and making the grid-year level ($0.1^\circ \times 0.1^\circ$) prediction at the global scale have become possible. Considering the availability of data, several global fine-scale datasets described in Methods section were used to generate the candidate independent variables. The candidate independent variables adopted in the present study were divided into two categories: climate deviation related factors and stable background contexts. Climate deviation related factors included: (a) Standardized temperature index (One-year or Two-year); (b) standardized precipitation index (One-year or Two-year). Stable background contexts included: (a) Mean temperature; (b) Mean precipitation; (c) Elevation; (d) Natural disaster hotspots; (e) Ethnic diversity; (f) Urban accessibility; (g) Nighttime lights; (h) Normalized difference vegetation index. The list of independent variables and statistical tests under different modelling strategies are detailed in chapters Modelling Strategy and Statistical Test respectively.” (Pages 4, lines 36-56 in Supplementary Information).

1) Sundberg R, Melander E. Introducing the UCDP georeferenced event dataset. Journal of

- Peace Research 50, 523-532 (2013).
- 2) Eck, K. In data we trust? A comparison of UCDP GED and ACLED conflict events datasets. *Cooperation & Conflict* 47, 124-141 (2012).
 - 3) Van Weezel, S. Economic shocks & civil conflict onset in Sub-Saharan Africa, 1981–2010. *Defence and Peace Economics* 26, 153-177 (2015).
 - 4) Uexkull VN, Croicu M, Fjelde H, Buhaug H. Civil conflict sensitivity to growing-season drought. *Proceedings of the National Academy of Sciences* 113, 12391-12396 (2016).
 - 5) Hsiang SM, Burke M, Miguel E. Quantifying the Influence of Climate on Human Conflict. *Science* 341, 1235367 (2013).
 - 6) Schleussner CF, Donges JF, Donner RV, Schellnhuber HJ. Armed-conflict risks enhanced by climate-related disasters in ethnically fractionalized countries. *Proceedings of the National Academy of Sciences* 113, 9216-9221 (2016).
 - 7) O’Loughlin J, Witmer FD, Linke AM, Laing A, Gettelman A, Dudhia J. Climate variability and conflict risk in East Africa, 1990–2009. *Proceedings of the National Academy of Sciences* 109, 18344-18349 (2012).
 - 8) Burke MB, Miguel E, Satyanath S, Dykema JA, Lobell DB. Warming increases the risk of civil war in Africa. *Proceedings of the National Academy of Sciences of the United States of America* 106, 20670-20674 (2009).
 - 9) Barnaby W. Do nations go to war over water? *Nature* 458, 282-283 (2009).
 - 10) O’Loughlin J, Linke AM, Witmer FD. Effects of temperature and precipitation variability on the risk of violence in sub-Saharan Africa, 1980–2012. *Proceedings of the National Academy of Sciences* 111, 16712-16717 (2014).
 - 11) Hegre H, et al. ViEWS: A political violence early-warning system. *Journal of Peace Research* 56, 155-174 (2019).
 - 12) Bazzi S, Blair R, Blattman C, Dube O, Gudgeon M, Peck R. The Promise and Pitfalls of Conflict Prediction: Evidence from Colombia and Indonesia. *The Review of Economics and Statistics*, 1-45 (2021).

Appendix to Response to Reviewer's comment No. 3

Table S6. Correlation matrix between covariate variables used in BRT ensembles trained on one-year (2000) onset samples under strategy a.

	SPI	STI	E	NTL	UA	ED	NDH	MP	MT	NDVI
SPI	1	-0.301	-0.148	-0.032	0.094	-0.012	0.095	0.24	0.016	0.146
STI	-0.301	1	0.058	-0.043	-0.017	0.064	-0.074	-0.371	-0.039	-0.213
E	-0.148	0.058	1	-0.114	0.029	0.112	0.006	-0.126	-0.156	-0.177
NTL	-0.032	-0.043	-0.114	1	-0.196	0.106	-0.008	-0.002	0.107	0.021
UA	0.094	-0.017	0.029	-0.196	1	-0.319	-0.14	-0.226	-0.539	-0.365
ED	-0.012	0.064	0.112	0.106	-0.319	1	0.137	0.129	0.212	0.193
NDH	0.095	-0.074	0.006	-0.008	-0.14	0.137	1	0.276	0.163	0.123
MP	0.24	-0.371	-0.126	-0.002	-0.226	0.129	0.276	1	0.495	0.703
MT	0.016	-0.039	-0.156	0.107	-0.539	0.212	0.163	0.495	1	0.485
NDVI	0.146	-0.213	-0.177	0.021	-0.365	0.193	0.123	0.703	0.485	1

Note: SPI (Standardized precipitation index): VIF = 1.170; STI (Standardized temperature index): VIF = 1.280; E (Elevation): VIF = 1.110; NTL (Nighttime lights): VIF = 1.074; UA (Urban accessibility): VIF = 1.648; ED (Ethnic diversity): VIF = 1.175; NDH (Natural disaster hotspots) : VIF = 1.125; MP (Mean precipitation) : VIF = 2.673; MT (Mean temperature) : VIF = 1.859; NDVI (normalized difference vegetation index) : VIF = 2.284.

Table S7. Correlation matrix between covariate variables used in BRT ensembles trained on one-year (2000) onset samples under strategy a+.

	SPI	STI	E	NTL	UA	ED	NDH	MP	MT	NDVI
SPI	1	-0.396	-0.131	-0.071	0.054	-0.007	0.131	0.364	0.137	0.23
STI	-0.396	1	0.084	0.036	-0.094	0.113	0.047	-0.348	-0.057	-0.264
E	-0.131	0.084	1	-0.114	0.029	0.112	0.006	-0.126	-0.156	-0.177
NTL	-0.071	0.036	-0.114	1	-0.196	0.106	-0.008	-0.002	0.107	0.021
UA	0.054	-0.094	0.029	-0.196	1	-0.319	-0.14	-0.226	-0.539	-0.365
ED	-0.007	0.113	0.112	0.106	-0.319	1	0.137	0.129	0.212	0.193
NDH	0.131	0.047	0.006	-0.008	-0.14	0.137	1	0.276	0.163	0.123
MP	0.364	-0.348	-0.126	-0.002	-0.226	0.129	0.276	1	0.495	0.703
MT	0.137	-0.057	-0.156	0.107	-0.539	0.212	0.163	0.495	1	0.485
NDVI	0.23	-0.264	-0.177	0.021	-0.365	0.193	0.123	0.703	0.485	1

Note: SPI (Standardized precipitation index): VIF = 1.317; STI (Standardized temperature index): VIF = 1.356; E (Elevation): VIF = 1.101; NTL (Nighttime lights): VIF = 1.070; UA (Urban accessibility): VIF = 1.664; ED (Ethnic diversity): VIF = 1.180; NDH (Natural disaster hotspots): VIF = 1.149; MP (Mean precipitation): VIF = 2.603; MT (Mean temperature): VIF = 1.829; NDVI (normalized difference vegetation index): VIF = 2.301.

Table S8. Correlation matrix between covariate variables used in BRT ensembles trained on one-year (2000) onset samples under strategy b.

	SPI	STI	MP	MT
SPI	1	-0.301	0.24	0.016
STI	-0.301	1	-0.371	-0.039

MP	0.24	-0.371	1	0.495
MT	0.016	-0.039	0.495	1

Note: SPI (Standardized precipitation index): VIF = 1.131; STI (Standardized temperature index): VIF = 1.258; MP (Mean precipitation): VIF = 1.630; MT (Mean temperature): VIF = 1.379.

Table S9. Correlation matrix between covariate variables used in BRT ensembles trained on one-year (2000) onset samples under strategy b+.

	SPI	STI	MP	MT
SPI	1	-0.396	0.364	0.137
STI	-0.396	1	-0.348	-0.057
MP	0.364	-0.348	1	0.495
MT	0.137	-0.057	0.495	1

Note: SPI (Standardized precipitation index): VIF = 1.274; STI (Standardized temperature index): VIF = 1.279; MP (Mean precipitation): VIF = 1.621; MT (Mean temperature): VIF = 1.352.

Table S14. Correlation matrix between covariate variables used in BRT ensembles trained on all onset samples under strategy a.

	SPI	STI	E	NTL	UA	ED	NDH	MP	MT	NDVI
SPI	1	-0.092	0.005	-0.027	0.013	0.012	0.03	0.006	-0.025	0.035
STI	-0.092	1	0.021	-0.027	-0.051	-0.03	-0.021	-0.025	0.219	0.03
E	0.005	0.021	1	-0.098	0.044	0.111	-0.017	-0.146	-0.152	-0.187
NTL	-0.027	-0.027	-0.098	1	-0.178	0.096	0.022	-0.003	0.099	0.029
UA	0.013	-0.051	0.044	-0.178	1	-0.279	-0.131	-0.139	-0.504	-0.262
ED	0.012	-0.03	0.111	0.096	-0.279	1	0.11	0.116	0.18	0.159
NDH	0.03	-0.021	-0.017	0.022	-0.131	0.11	1	0.304	0.172	0.171
MP	0.006	-0.025	-0.146	-0.003	-0.139	0.116	0.304	1	0.425	0.727
MT	-0.025	0.219	-0.152	0.099	-0.504	0.18	0.172	0.425	1	0.376
NDVI	0.035	0.03	-0.187	0.029	-0.262	0.159	0.171	0.727	0.376	1

Note: SPI (Standardized precipitation index): VIF = 1.014; STI (Standardized temperature index): VIF = 1.099; E (Elevation): VIF = 1.089; NTL (Nighttime lights): VIF = 1.050; UA (Urban accessibility): VIF = 1.530; ED (Ethnic diversity): VIF = 1.133; NDH (Natural disaster hotspots) : VIF = 1.129; MP (Mean precipitation) : VIF = 2.590; MT (Mean temperature) : VIF = 1.765; NDVI (normalized difference vegetation index) : VIF = 2.339.

Table S15. Correlation matrix between covariate variables used in BRT ensembles trained on all onset samples under strategy a+.

	SPI	STI	E	NTL	UA	ED	NDH	MP	MT	NDVI
SPI	1	-0.073	-0.005	-0.042	0.02	0.007	0.05	0.045	-0.01	0.069
STI	-0.073	1	0.028	-0.017	-0.056	-0.031	-0.025	-0.042	0.237	0.02
E	-0.005	0.028	1	-0.098	0.044	0.111	-0.017	-0.146	-0.152	-0.187
NTL	-0.042	-0.017	-0.098	1	-0.178	0.096	0.022	-0.003	0.099	0.029
UA	0.02	-0.056	0.044	-0.178	1	-0.279	-0.131	-0.139	-0.504	-0.262
ED	0.007	-0.031	0.111	0.096	-0.279	1	0.11	0.116	0.18	0.159
NDH	0.05	-0.025	-0.017	0.022	-0.131	0.11	1	0.304	0.172	0.171

MP	0.045	-0.042	-0.146	-0.003	-0.139	0.116	0.304	1	0.425	0.727
MT	-0.01	0.237	-0.152	0.099	-0.504	0.18	0.172	0.425	1	0.376
NDVI	0.069	0.02	-0.187	0.029	-0.262	0.159	0.171	0.727	0.376	1

Note: SPI (Standardized precipitation index): VIF = 1.016; STI (Standardized temperature index): VIF = 1.119; E (Elevation): VIF = 1.090; NTL (Nighttime lights): VIF = 1.051; UA (Urban accessibility): VIF = 1.533; ED (Ethnic diversity): VIF = 1.133; NDH (Natural disaster hotspots): VIF = 1.130; MP (Mean precipitation): VIF = 2.606; MT (Mean temperature): VIF = 1.799; NDVI (normalized difference vegetation index): VIF = 2.344.

Table S16. Correlation matrix between covariate variables used in BRT ensembles trained on all onset samples under strategy b.

	SPI	STI	MP	MT
SPI	1	-0.092	0.006	-0.025
STI	-0.092	1	-0.025	0.219
MP	0.006	-0.025	1	0.425
MT	-0.025	0.219	0.425	1

Note: SPI (Standardized precipitation index): VIF = 1.009; STI (Standardized temperature index): VIF = 1.077; MP (Mean precipitation): VIF = 1.242; MT (Mean temperature): VIF = 1.304.

Table S17. Correlation matrix between covariate variables used in BRT ensembles trained on all onset samples under strategy b+.

	SPI	STI	MP	MT
SPI	1	-0.073	0.045	-0.01
STI	-0.073	1	-0.042	0.237
MP	0.045	-0.042	1	0.425
MT	-0.01	0.237	0.425	1

Note: SPI (Standardized precipitation index): VIF = 1.007; STI (Standardized temperature index): VIF = 1.093; MP (Mean precipitation): VIF = 1.256; MT (Mean temperature): VIF = 1.326.

Supplementary Figure 4. Validation performance on a time scale of the BRT models trained on one-year onset samples. Validation performance of strategies a and a+ are shown in the left and right columns, respectively.

Supplementary Figure 5. Validation performance on a time scale of the BRT models trained on one-year onset samples. Validation performance of strategies b and b+ are shown in the left and right columns, respectively.

Supplementary Figure 9. Marginal effect curves of each stable background covariate over the BRT ensembles fitted to the full onset samples under strategy a+. The black lines represent the mean effect curves calculated from the ensemble BRT models and the dark grey the 95% confidence interval. Sub-plots are ordered by the mean relative contribution (%) of covariates, with these mean relative contribution \pm standard deviation (%) given within each sub-plot.

Supplementary Figure 10. Marginal effect curves of each stable background covariate over the BRT ensembles fitted to the full onset samples under strategy a. The black lines represent the mean effect curves calculated from the ensemble BRT models and the dark grey the 95% confidence interval. Sub-plots are ordered by the mean relative contribution (%) of covariates, with these mean relative contribution \pm standard deviation (%) given within each sub-plot.

Supplementary Figure 11. Marginal effect curves of each climate deviation related covariate over the BRT ensembles fitted to the full onset samples under strategy a+. The white lines represent the mean effect curves calculated from the ensemble BRT models. 95% confidence interval of climate variables are indicated by color: red, standardized temperature index; blue, Standardized precipitation index. Sub-plots are ordered by the mean relative contribution (%) of covariates, with these mean relative contribution \pm standard deviation (%) given within each sub-plot.

Supplementary Figure 12. Marginal effect curves of each climate deviation related covariate over the BRT ensembles fitted to the full onset samples under strategy a. The white lines represent the mean effect curves calculated from the ensemble BRT models. 95% confidence interval of climate variables are indicated by color: red, standardized temperature index; blue, Standardized precipitation index. Sub-plots are ordered by the mean relative contribution (%) of covariates, with these mean relative contribution \pm standard deviation (%) given within each sub-plot.

Supplementary Figure 17. Maps of the global simulated risk of armed conflict onset at $0.1^\circ \times 0.1^\circ$ spatial resolution based on 20 ensemble BRT models trained on all onset samples under strategy a. The simulated risk level ranges from 0 (blue) to 1 (red).

Supplementary Figure 18. Maps of the global simulated risk of armed conflict onset at $0.1^\circ \times 0.1^\circ$ spatial resolution based on 20 ensemble BRT models trained on all onset samples under strategy a+. The simulated risk level ranges from 0 (blue) to 1 (red).

Supplementary Figure 19. Maps of the global simulated risk of armed conflict onset at $0.1^\circ \times 0.1^\circ$ spatial resolution based on 20 ensemble BRT models trained on all onset samples under strategy b. The simulated risk level ranges from 0 (blue) to 1 (red).

Supplementary Figure 20. Maps of the global simulated risk of armed conflict onset at $0.1^\circ \times 0.1^\circ$ spatial resolution based on 20 ensemble BRT models trained on all onset samples under strategy b+. The simulated risk level ranges from 0 (blue) to 1 (red).

Supplementary Figure 25. Maps of uncertainty associated with these simulations derived from 20 ensemble BRT models trained on all onset samples under strategy a.

Supplementary Figure 26. Maps of uncertainty associated with these simulations derived from 20 ensemble BRT models trained on all onset samples under strategy a+.

Supplementary Figure 27. Maps of uncertainty associated with these simulations derived from 20 ensemble BRT models trained on all onset samples under strategy b.

Supplementary Figure 28. Maps of uncertainty associated with these simulations derived from 20 ensemble BRT models trained on all onset samples under strategy b+.

Supplementary Table 2. The performance of the 20 ensemble BRT models trained on one-year onset samples during time-cross validation process.

Performance	Strategy a		Strategy a+		Strategy b		Strategy b+	
	Mean	Standard Deviation	Mean	Standard Deviation	Mean	Standard Deviation	Mean	Standard Deviation
ROC-AUC	0.873	0.036	0.880	0.035	0.785	0.056	0.798	0.054
PR-AUC	0.842	0.045	0.851	0.044	0.731	0.071	0.749	0.068
F1-score	0.762	0.067	0.771	0.066	0.661	0.090	0.677	0.086

Supplementary Table 4. The performance of the 20 ensemble BRT models trained on all onset samples under different strategies.

Performance	Strategy a		Strategy a+		Strategy b		Strategy b+	
	Mean	Standard Deviation	Mean	Standard Deviation	Mean	Standard Deviation	Mean	Standard Deviation
ROC-AUC	0.927	0.002	0.928	0.002	0.868	0.004	0.874	0.004
PR-AUC	0.926	0.002	0.928	0.002	0.862	0.004	0.869	0.004
F1-score	0.872	0.002	0.874	0.002	0.820	0.004	0.823	0.004

Supplementary Table 6. The significant differences that were observed for the ROC-AUC performance of the 20 ensemble BRT models trained on all onset samples under different strategies.

Strategy	a	a+	b	b+
a	—	—	—	—
a+	0.015 *	—	—	—
b	1.451E-11 ***	6.786E-8 ***	—	—
b+	1.451E-11 ***	6.786E-8 ***	2.898E-5 ***	—

Note: * indicates $p < 0.05$; ** indicates $p < 0.01$; *** indicates $p < 0.001$; NS indicates not significant; The p values were determined by two-tailed Mann–Whitney test, representing a comparison among strategies.

Supplementary Table 8. The relative contribution of covariates in simulating the global risk of armed conflict onset based on the 20 ensemble BRT models trained on all onset samples from period 2000-2015 under strategies a and a+.

Variables	Relative contribution \pm Standard Deviation, %	
	Strategy a	Strategy a+
stable background covariates [†]	96.894	96.067
Mean temperature	47.763 \pm 1.667	47.158 \pm 1.589
Natural disaster hotspots	14.267 \pm 1.239	14.037 \pm 1.206
Mean precipitation	10.896 \pm 1.052	10.902 \pm 1.057
Urban accessibility	9.901 \pm 0.933	9.830 \pm 0.928
Elevation	5.587 \pm 0.370	5.293 \pm 0.360
Ethnic diversity	3.093 \pm 0.240	3.045 \pm 0.214
Nighttime lights	2.815 \pm 0.239	2.846 \pm 0.244
Normalized difference vegetation index	2.572 \pm 0.284	2.557 \pm 0.272
climate deviations related covariates [†]	3.106	4.331
Standardized temperature index	2.233 \pm 0.314	2.955 \pm 0.341
Standardized precipitation index	0.873 \pm 0.092	1.376 \pm 0.153

Note: [†]Sum of relative contribution for both categories.

REVIEWER COMMENTS

Reviewer #4 (Remarks to the Author):

This is somewhat improved but still problematic. There are major nomenclature errors or confusion.

Consider the abstract, which says that " The inferred patterns show that positive temperature deviation or high precipitation extremes are associated with increased risk of armed conflicts worldwide." Conflicts here remains ambiguous, and technically incorrect.

I think this is simple: almost every time the authors write "conflicts" they mean "conflict events" and they should correct all of these instances. This is true in almost every paragraph.

This is a critical distinction in the conflict literature. For example, if two rivals fight for 10 years, pause and fight for 2 more, and have 1000 battles in all, I believe that in the dataset that the authors use this corresponds to:

- 1 conflict
- 2 conflict episodes (some would say 2 conflicts, which would also be fine)
- 2 conflict onsets
- 12 conflict years (or years of conflict incidence)
- 1000 conflict events

The authors will want to double check these definitions in the conflict dataset, but I believe they are roughly correct.

All five are distinct concepts, with distinct theoretical determinants and predictors, and they should not be confused.

Responses to reviewers

NCOMMS-20-07482D

Title: Modelling armed conflict risk under climate change with machine learning and high-frequency time-series data

We sincerely thank all the reviewers for their critical reading and constructive comments and suggestions for improving our manuscript. We have carefully addressed all the questions mentioned in comments and revised the terminology, as requested. Answers are typed in **blue**. Following the reviewers' comments and suggestions, we have revised the manuscript accordingly, as detailed below.

Reviewer #4 (Remarks to the Author):

Point No.1 This is somewhat improved but still problematic. There are major nomenclature errors or confusion.

Consider the abstract, which says: "The inferred patterns show that positive temperature deviation or high precipitation extremes are associated with increased risk of armed conflicts worldwide." Conflicts here remains ambiguous, and technically incorrect.

I think this is simple: almost every time the authors write "conflicts" they mean "conflict events" and they should correct all of these instances. This is true in almost every paragraph.

This is a critical distinction in the conflict literature. For example, if two rivals fight for 10 years, pause and fight for 2 more, and have 1000 battles in all, I believe that in the dataset that the authors use this corresponds to:

- 1 conflict
- 2 conflict episodes (some would say 2 conflicts, which would also be fine)
- 2 conflict onsets
- 12 conflict years (or years of conflict incidence)
- 1000 conflict events

The authors will want to double check these definitions in the conflict dataset, but I believe they are roughly correct. All five are distinct concepts, with distinct theoretical determinants and predictors, and they should not be confused.

Response to Reviewer’s comment No. 1: We are very grateful to you for pointing out the unclear nomenclature and providing suggestions to improve our manuscript. According to your valuable advice, we rechecked the corresponding terms and improved the ambiguous “conflict” terminology used in this study. We fully agree that the used technical terms (i.e., conflict, conflict events and armed conflict onset or incidence) are distinct concepts, with distinct theoretical determinants and predictors, and they should not be confused. Where meaningful we used the term “armed conflict events”, except where we referred to conflict in general terms or cited other sources.

In the revised version, we define the technical terms in the first paragraph of the introduction section to make the terms clear and avoid confusion. The details are as follows:

“Among various conceptions of armed conflict most prominent is the Uppsala Conflict Data Program (UCDP) georeferenced event dataset (GED) which defines an armed conflict event as “An incident where armed force was used by an organised actor against another organized actor, or against civilians, resulting in at least 1 direct death at a specific location and a specific date” (ref. 3). This allows to measure the frequency of armed conflict events in terms of incidence (armed conflict event in a particular year) and onset (incidence without armed conflict event in previous year) in spatial and time units (see the equations in Supplementary Information). In our analysis, we count the existence of both incidence and onset of armed conflict events, while other aspects of armed conflict are not specified such as conflict intensity or consequences, conflict parties, historical context or other patterns of conflict which are considered in the literature.”
(Pages 2-3, Lines 46-58)

Then, we further clarify the differences between conflict risk, armed conflict events, armed conflict incidence and armed conflict onset in the third paragraph of the introduction section. The details are as follows:

“In recent years, understanding conflict risk have drawn increased attention from an interdisciplinary group of scientists because it is of great significance for human safety and security (ref. 8). The term conflict risk has been associated with the probability of armed conflict events (ref. 9) which is adapted here to refer to the frequency of armed conflict events which involves armed conflict incidence and armed conflict onset. Both researchers and policy makers have recently discussed intensively whether climate change impacts conflict risks (refs. 9 and 10).

The United Nations Security Council, for instance, has conducted discussion on climate change and security in every year since 2018.” (Pages 3-4, Lines 69-78)

Accordingly, in the revised abstract section, we rephrased the corresponding technical terms in order to make the terminology clear. The details are as follows:

“Understanding the risk of armed conflict is essential for promoting peace. Although the relationship between climate variability and armed conflict has been studied by the research community for decades with quantitative and qualitative methods at different spatial and temporal scales, causal linkages at a global scale remain poorly understood. Here we adopt a quantitative modelling framework based on machine learning to infer potential causal linkages from high-frequency time-series data and simulate the risk of armed conflict worldwide from 2000-2015. Our results reveal that the risk of armed conflict is primarily influenced by stable background contexts with complex patterns, followed by climate deviations related covariates. The inferred patterns show that positive temperature deviations or precipitation extremes are associated with increased risk of armed conflict worldwide. Our findings indicate that a better understanding of climate-conflict linkages at the global scale enhances the spatiotemporal modelling capacity for the risk of armed conflict.” (Page 2, Lines 26-41)

In addition, the title of the manuscript was also revised, as following:

“Modelling armed conflict risk under climate change with machine learning and high-frequency time-series data” (Page 1, Lines 2-3)

In addition, we also rephrased the corresponding statements throughout this revised manuscript and the revised supplementary materials. For instance:

“According to UCDP-GED, more than 91,000 armed conflict events occurred globally between 2000 and 2015, which caused approximately 654,000 deaths, including nearly 144,000 civilians (ref. 4). In Asia and Africa, the Armed Conflict Location and Event Data Project (ACLED) reported that more than 23,000 armed conflict events occurred from January to August 2017, killing approximately 24,000 people (ref. 5). Although the global trend of armed conflict events has declined in both number and intensity over a decade long perspective, with particularly sharp declines in higher-intensity conflict (refs. 6 and 7), the frequency of armed conflict events in several areas shows an upward trend, becoming more concentrated in Africa, the Middle East and South Asia (ref. 5).” (Page 3, Lines 59-68)